# A comprehensive global dataset of atmospheric [7]Be and [210]Pb measurements: air concentration and depositional flux

Fule Zhang[1], Jinlong Wang[1], Mark Baskaran[2], Qiangqiang Zhong[3], Yali Wang[1], Jussi Paatero[4], Jinzhou Du[1].

[1]State Key Laboratory of Estuarine and Coastal Research, East China Normal University, Shanghai, 200241, China

[2]Department of Environmental Science and Geology, Wayne State University, Detroit, Michigan 48202, USA

[3]Laboratory of Marine Isotopic Technology and Environmental Risk Assessment, Third Institute of Oceanography, Ministry of Natural Resource, Xiamen, 361005, China

[4]Observation Services, Finnish Meteorological Institute, Helsinki, FI-00560, Finland

*Correspondence to*: Jinlong Wang (jlwang@sklec.ecnu.edu.cn)

**Abstract.** [7]Be and [210]Pb air concentration and depositional flux data provide key information on the origins and movements of air masses, as well as deposition processes and residence time of aerosols. After their deposition onto the Earth's surface, they are utilized for tracing soil redistribution processes on land and particle dynamics and mixing processes in the ocean. Here we present a global dataset of air concentration and depositional flux measurements of atmospheric [7]Be and [210]Pb made by a large number of researchers and laboratories. Data were collected from published papers between 1955 and early 2020. It includes the annual surface air concentrations data of [7]Be from 367 sites and of [210]Pb from 270 sites, the annual depositional flux of [7]Be from 279 sites, and of [210]Pb from 602 sites. When available, appropriate metadata have also been summarized, including geographic location, sampling date, methodology, annual precipitation, and references. The dataset is available at https://doi.org/10.5281/zenodo.4521649 (Zhang et al., 2021). The purpose of this paper is to have the published data available in one place for future researchers' public consumption in their research and provide an overview of the scope and nature of this dataset holdings.

## 1 Introduction

Naturally occurring beryllium-7 ([7]Be, $T_{1/2}$: 53.3 days) and lead-210 ([210]Pb, $T_{1/2}$: 22.3 y) have been widely utilized as tracers to investigate Earth's surface processes (Du et al., 2012). [7]Be, a cosmogenic



radionuclide, is produced by the spallation of oxygen and nitrogen nuclei by cosmic rays in the stratosphere and upper troposphere (Lal et al., 1958). Although a major fraction of $^7$Be (67%) production takes place in the stratosphere, they do not readily reach the troposphere except when tropopause folds near the jet stream seasonal thinning at mid-latitudes and changing the seasonal structure of the stratosphere facilitating the exchange of air masses (Lal and Peters, 1967; Danielsen, 1968). Depositional

flux of $^7$Be is independent of longitude but depends on the altitude and the ~11 years solar cycle. Besides, owing to the tropospheric/stratospheric exchange occurring in the mid-latitudes during spring, $^7$Be flux to the Earth's surface varies with latitude and season (Lal and peters, 1967; Koch and Mann, 1996).$^{210}$Pb, a progeny of $^{222}$Rn in the $^{238}$U-series, is derived mostly (>99%) from the radioactive decay of $^{222}$Rn. Most of the atmospheric $^{210}$Pb is derived from atmospheric radon. The global $^{222}$Rn flux from continent ranges

from 1,300 to 1,800 Bq m$^{-2}$ d$^{-1}$ while ~17 Bq m$^{-2}$ d$^{-1}$ is reported for the oceanic areas (Samuelsson et al., 1986; Nazaroff, 1992). Radon is not removed by precipitation scavenging while its decay products are. Vertical profiles of $^{222}$Rn in the atmosphere indicate the highest concentrations in the continental boundary layer (CBL, 3-8 Bq m$^{-3}$) and an order of magnitude lower activity (~40 mBq m$^{-3}$) near the tropopause, with decreasing activity with increasing altitude from CBL (Moore et al., 1977; Liu et al.,

1984; Kritz et al., 1993). Consequently, the atmospheric concentration of $^{210}$Pb decreases with altitude and is strongly controlled by the land-sea distribution pattern. After formation, both $^7$Be and $^{210}$Pb or its predecessors, short-lived $^{222}$Rn progeny, are rapidly and irreversibly attached to aerosol particles (Winkler et al., 1998; Elsässer et al., 2011). Subsequently, the fate of $^7$Be and $^{210}$Pb is closely linked to that of aerosols. Most of $^7$Be and $^{210}$Pb are in accumulation mode aerosol particles with an aerodynamic

diameter of a few hundred nanometers (Ioannidou and Paatero, 2014; Paatero et al., 2017). Therefore, they are deposited onto the Earth's surface primarily by precipitation because accumulation mode aerosol particles are too small for gravitational settling and removal and too large to be deposited by Brownian motion.

     Owing to their distinctly different source terms but well-known source distributions, similar tropospheric

physicochemical behavior, $^7$Be and $^{210}$Pb have been widely utilized as powerful atmospheric tracers for study the origin of the air masses (e.g. Graustein and Turekian, 1996; Zheng et al., 2005; Likuku et al., 2006; Dueñas et al., 2011; Lozano et al., 2012), vertical exchange and horizontal transport processes (e.g. Arimoto et al., 1999; Lee et al., 2007; Rastogi and Sarin, 2008; Tositti et al., 2014), deposition velocities and washout ratios of aerosols (e.g. Todd et al., 1989; McNeary and Baskaran, 2003, Dueñas et al., 2005;



Lozano et al., 2011; Mohan et al., 2019) and behavior and fate of analog species (e.g. Crecelius, 1981; Mattson, 1988; Prospero et al., 1995; Lamborg et al., 2013). Following their deposition on the Earth's surface, both $^7$Be and $^{210}$Pb are strongly attached to soils, which make them useful for assessing soil erosion rates from episodic to multi-decadal timescales (e.g. Wallbrink and Murry, 1993; Walling and He, 1999; Blake et al., 1999; Walling et al., 1999; Wilson et al., 2003; Mabit et al., 2008, 2014). In aqueous

environments, $^{210}$Pb is most widely used for dating recent sediments (Appleby, 2008). Meanwhile, $^7$Be and $^{210}$Pb are also widely used for indicating particle transport, deposition, and resuspension in estuarine and coastal regions (e.g. Du et al., 2010; Huang et al., 2013; Wang et al., 2016). $^7$Be deposited on the open ocean are further used as a tracer for diagnosing ocean ventilation and subduction (Kadko, 2000; Kadko and Olson, 1996), inferring upwelling rates (Kadko and Johns, 2011) and estimating the

deposition of trace metals (Kadko et al 2015; Shelley et al., 2017; Buck et al., 2019).

There have been numerous published datasets with concentrations and depositional fluxes of $^{210}$Pb and $^7$Be (directly and indirectly) over the past few decades, particularly under the national or international monitoring programs such as Environmental Measurements Laboratory (EML) Surface Air Sampling Program (Feely et al., 1989), Sea-Air Exchange (SEAREX) program (Uematsu et al., 1994), Finnish

Meteorological Institute monitoring program (Paatero et al., 2015), Radioactivity Environmental Monitoring (REM) network (Hernandez-Ceballos et al., 2015), and International Monitoring System (IMS) by Comprehensive Nuclear-Test-Ban Treaty Organization (CTBTO) (Terzi and Kalinowski, 2017). It is valuable to compile all these existing data, including those measured in case studies, along with appropriate metadata, in one place for facilitating further analysis and application.

Although several datasets of air concentrations and depositional fluxes $^7$Be (Bleichrodt, 1978; Brost et al., 1991) and $^{210}$Pb (Rangarajan et al., 1986; Preiss et al., 1996) have been published, unfortunately, many of the published data are not readily available in hard data format (e.g, data table) and often it is challenging and not precise to retrieve data from published figures. To date, only one dataset was published that compiled $^7$Be and $^{210}$Pb together (Persson, 2016), but it contained limited data. Therefore,

the focus of the present work is to build a new and comprehensive dataset. This dataset is the result of many scientists' efforts in generating the data.

**2 Methods**

**2.1 $^7$Be and $^{210}$Pb air concentrations measurement methodology**

Aerosol samples are usually collected by pumping high volume air, typically 1.4-1.5 m³/min, through paper filters at a relatively constant flow rate. Common aerosol collecting equipment includes ASS 500 (CLRP Warsaw, Germany), Anderson PM10 (Anderson Ltd., USA), Snow White (Senya Ltd., Finland), and HV-1000F (Shibata Co. Ltd., Japan). The preferred filter membrane material includes glass fiber, cellulose nitrate or acetate, polypropylene fiber, and quartz fiber. The collection efficiency, the percentage of the particles in the air stream that are collected by the air filter, depends on the aerodynamic diameter

of aerosol particles and the filter face velocity (average flow velocity of air into the filter) of the airflow. Although the collection efficiency depends on the distribution of particle sizes, generally the collection efficiency varies between 80% and 100% for different filter materials. This was shown by overlapping two filters in tandem and comparing the concentrations separately in the top and bottom filters. The sampling frequency is usually set from daily to monthly, with a typical collection time of ~ 24 h,

corresponding to ~ 2,000 m³ air. The ⁷Be and ²¹⁰Pb trapped on filters can both be analyzed simultaneously by gamma spectroscopy (e.g. Bourcier et al., 2011; Lozano et al., 2012; Mohan et al., 2018), while ²¹⁰Pb also can be analyzed by beta counting of its daughter ²¹⁰Bi (e.g. Joshi et al., 1969; Poet et al., 1972; Daish et al., 2005) or via alpha counting of in-grown ²¹⁰Po from the decay of ²¹⁰Pb (e.g. Turekian and Cochran, 1981; Mattsson et al., 1996; Marx et al., 2005).

**2.2 ⁷Be and ²¹⁰Pb depositional fluxes measurement methodology**

The atmospheric depositional fluxes of ⁷Be and ²¹⁰Pb are commonly measured directly by using rain collectors such as polyethylene drum/buckets, stainless steel container, and indirectly by natural archives (e.g. soils, lichens, mosses, snow/ice cores, salt marsh sediments). The direct collecting method is the most reliable technique for the measurement of annual ⁷Be and ²¹⁰Pb depositional fluxes, and this

technique is useful in collecting short-time scale (daily, weekly and monthly) depositional fluxes. In contrast, using natural archives the tedious procedures and can serve as a complementary to fill regional gaps especially in deserted areas. However, these archives are susceptible to be affected by natural processes and anthropogenic activities, thus, the sampling location of these archives should be restricted to an undisturbed area.

**2.2.1 Direct ⁷Be and ²¹⁰Pb flux measurements**

Rain collectors are usually placed on the roof of a building so as to prevent contamination from

resuspended dust from the ground. Care should be taken to ensure direct overhead atmospheric deposition is collected and there is no shadowing effect from adjacent structure/building or funneling effect in sample collection. Atmospheric aerosols can be removed not only by precipitation-scavenging but also

by settling under the influence of gravitational or electrostatic forces. In most cases, the collectors are continuously exposed over a long enough period and the bulk (wet + dry) depositional samples are collected periodically (e.g. Baskaran et al., 1993; Hirose et al., 2004; Baskaran and Swarzenski, 2007; Lozano et al., 2011; Du et al., 2015). Fluxes obtained by this method yields the best estimate of the depositional flux. Sometimes, fallout $^7$Be and $^{210}$Pb samples are collected only during rainfall, and

concentration is measured in the individual rainwater sample (e.g. Cho et al., 2011; Chae and Kim, 2019; Du et al., 2020). In this case, only the bulk depositional flux is obtained for the duration of collection. In rare cases, only a mean concentration of $^{210}$Pb and/or $^7$Be in rainwater is available, the wet flux can be estimated by multiplying by the annual precipitation (Peirson et al., 1966).

Most of the time, the volume of rainwater sample is large enough which cannot be directly counted in a

gamma-ray spectrometer for the simultaneous measurements of $^7$Be and $^{210}$Pb, and a preconcentration of the sample is required. Since both $^7$Be and $^{210}$Pb have a strong affinity for solid surfaces, it is strongly recommended to add stable Be (commonly 1-5 mg) and stable Pb (typically 5-20 mg) in about 1 L of 1 M HCl to the rain collector prior to deployment. Alternately, the spikes can also be immediately after sample collection, followed by rinsing of rain collector with 1 L of 1 M HCl rinsing twice and combining

the rinses with the collected rainwater. An earlier critical review of earlier atmospheric depositional flux studies by Lal et al. (1979) showed that loss of $^7$Be and $^{210}$Pb by sorption onto rain collector walls was observed when pre-acidification of the collector was not done resulting in underestimate of depositional flux. In the case of preconcentration by ferric chloride precipitation method, due to variable scavenging efficiency of $^7$Be and $^{210}$Pb during the preconcentration method, it is required to add stable Pb and Be as

yield tracer (e.g. Baskaran et al., 1993). The best chemical procedure to obtain high-quality data is to add acid and stable Be and Pb careers with 1 L of 1 M HCl to the rain collector prior to the start of the sample collection. The final calculation for depositional fluxes would involve chemical yield for $^7$Be and $^{210}$Pb, and appropriate decay corrections, as outlined in Baskaran et al. (1993) and Du et al. (2015).

### 2.2.2 Indirect $^7$Be flux measurements

Measurements of $^7$Be inventory in the upper oceanic water column, from air-sea interface until the layer



where [7]Be activity is below the detection limit (in >400 L water sample), can indirectly yield the bulk

depositional flux of [7]Be (Brost et al., 1991). This requires precise determination of the penetration depth

of [7]Be in the water column. The only uncertainty is the loss of [7]Be-laden sinking particles from the upper

water column where [7]Be is present. After deposited on the ocean surface, [7]Be is generally mixed

uniformly within the surface mixed layer (Young and Silker, 1980; Kadko and Olson, 1996). In Open

Ocean, the particle concentration is generally low, and a major fraction of [7]Be is expected to be in the

dissolved phase, thus allowing particle scavenging losses to be ignored (Silker, 1972; Andrews et al.,

2008). Therefore, in the absence of physical removal processes other than radioactive decay, the input

flux of [7]Be should be balanced by the [7]Be inventory integrated over the water column. In other words,

[7]Be flux of atmospheric fallout (Bq m$^{-2}$ d$^{-1}$) can be obtained from the [7]Be water column inventory (Bq

m$^{-2}$) multiplied by the decay constant (0.013 d$^{-1}$) of [7]Be. This method has been proven to be reliable in

open ocean due to the relatively short half-life of [7]Be and the constancy of [7]Be deposition over broad

latitudinal bands (Young and Silker, 1980; Aaboe et al., 1981). Although in areas where there are large

variations in seasonal variations of precipitation, the inventory is expected to be season-dependent (e.g.

monsoon-dominated areas in the global oceans), from the time-series study in Bermuda, the inventory of

[7]Be was relatively constant throughout the year, such that [7]Be inventory measured at any one time is

representative (to within 20%) of the instantaneous [7]Be flux (Kadko and Prospeo, 2011; Kodko et al.,

2015). This method is not suitable for coastal and estuarine areas where [7]Be is scavenged substantially

by particulate matters (Olsen et al., 1986; Baskaran and Santschi, 1993), and upwelling-dominated areas

where [7]Be inventory is diluted by [7]Be "dead" water (Kadko and Johns, 2011; Haskell et al., 2015).

Another potential candidate is undisturbed soil profiles. However, [7]Be inventories in undisturbed soils

were reported to vary by more than an order of magnitude within one year (Walling et al., 2009; Kaste et

al., 2011; Zhang et al., 2013). Such large variations in depositional fluxes of [7]Be are attributed to the

seasonal fluxes of [7]Be. Earlier studies have shown the atmospheric fluxes are highly dependent upon the

amount of precipitation (Baskaran, 1995; Du et al., 2015). Since seasonal variations will significantly

affect the [7]Be inventory in soils, and hence those are data are not included.

**2.2.3 Indirect [210]Pb flux measurements**

Several archives (soils, snow/ice cores, sediment cores, etc) have been used to assay the [210]Pb fluxes.

Here we only present [210]Pb fluxes estimated from soil profiles and snow/ice cores. The former is the most



frequently used, the latter perfectly fills the regional gap in polar regions and montane permanent
        snowfields. There are many $^{210}$Pb measurements in sediment cores, however, due to the sediment
        focusing and erosion, most sediment cores do not provide a reliable estimate of the atmospheric $^{210}$Pb
        flux (Turekian et al., 1977; Preiss et al., 1996), thus this type of $^{210}$Pb depositional flux data are not
        included in our dataset.

$^{210}$Pb in surface (upper ~30 cm) soil has two sources: one is generated from the decay of $^{222}$Rn in the soil
        minerals, known as supported $^{210}$Pb which is produced from the decay of $^{238}$U and the other comes from
        atmospheric deposition as unsupported $^{210}$Pb. The fallout of $^{210}$Pb is retained generally in the organic-
        rich surface soils presumably because of the sequestering properties of the organic matter as well as in
        lithogenic mineral grain. When soil $CO_2$ combines with percolating water, carbonic acid is produced

which can leach of the sorbed $^{210}$Pb ultimately resulting in slow migration down to a maximum depth of
        20-30 cm (Matisoff and Whiting, 2012). As a result, the surface soil layer contains an excess $^{210}$Pb
        concentration than that expected from its equilibrium with $^{226}$Ra (Mabit et al., 2014). The part of the $^{210}$Pb
        excessing is termed "unsupported" or "excess" $^{210}$Pb ($^{210}$Pb$_{ex}$). The mean residence time of $^{210}$Pb over a
        large drainage basin is on the order of 2000-3000 years in surface soils (Benninger et al., 1975; Dominik

et al., 1987), so the inventory of $^{210}$Pb$_{ex}$ in a soil profile that has not been disturbed by erosion,
        accumulation or human activities for about a century can be used to calculate the depositional flux
        (Graustein and Turekian, 1986). At a steady state, the $^{210}$Pb depositional flux can be deduced using the
        $^{210}$Pb$_{ex}$ inventory multiplied by the decay constant (0.0311 y$^{-1}$) of $^{210}$Pb. This method has been widely
        used worldwide (e.g. Nozaki et al., 1978; Graustein and Turekian, 1986, 1989; Dörr and Munich, 1991;

García-Orellana et al., 2006). At undisturbed soil sites, flux values derived from soil profile
        measurements were consistent with direct atmospheric flux observations (Olsen et al., 1985; Appleby et
        al., 2002, 2003), and $^{210}$Pb$_{ex}$ soil inventory showed little discrepancy at different sampling time (Porto et
        al., 2006, 2016).

        Goldberg (1963) was the first to show that the total $^{210}$Pb activity in a glacier from Greenland decreased

with depth, with a possibility of the dating ice core. Subsequently, snow chronology in the Antarctic was
        determined (Crozaz et al. 1964; Picciotto et al., 1964). Since then, this technique has been used in both
        the large ice caps of Antarctica (e.g. Picciotto et al., 1968; Koide et al., 1979; Nijampurkar et al., 2002)
        and Arctic (e.g. Crozaz and Langway, 1966; Koide et al., 1977; Dibb et al., 1990a, 1992; Peters et al.,
        1997) and the small montane permanent snow filed (e.g. Windom, 1969; Gaggeler et al., 1983; Monaghan

and Holdsworth, 1990). The $^{210}$Pb flux in snow/ice core is calculated in the same way as the soil, except the difference is that supported $^{210}$Pb in snow/ice core is very low due to low concentration of $^{226}$Ra in snow/ice core and may be negligible, as the lithogenic dust is the primary source of $^{226}$Ra and its concentration in polar regions are very low (Preiss et al., 1996). When the snow accumulation rate is known, the depositional flux can be also obtained by using $^{210}$Pb concentration in surface snow multiplied

by the accumulation rate (Pourchet et al., 1997; Suzuki et al., 2004). The uncertainty in the depositional flux of $^{210}$Pb from snow/ice core record is the potential post-depositional movement of the snow/ice due to heavy wind and the possibility of snow melting and percolation.

**2.3 Data collection**

Regarding compiling the global dataset for annual $^7$Be and $^{210}$Pb air concentrations and depositional

fluxes, we attempted to collect published papers between 1955 and early 2020 in which hard data for their concentrations and depositional fluxes are available or their calculated values reported. Using a series of keywords or with a combination of words search (e.g. $^7$Be, $^{210}$Pb, air concentration, depositional flux, fallout radionuclide, atmospheric tracer, or soil erosion), data were retrieved from online literature databases (Web of Science, Science Direct, and China Knowledge Resource Integrated Database).

During the literature survey, no a priori criteria (e.g. study area, sampling period, and measurement method) were applied. However, a critical review of the collected literature was conducted to obtain long-term data, using the following criteria. For concentrations in air and directly measured fluxes of $^7$Be and $^{210}$Pb, only those sites where more than one year of data were included. When averaged over a longer period, data are more representative because of the inherent seasonal variations (at least one full year

data) and inter-annual fluctuations (multi-year data). For indirectly measured fluxes of $^7$Be and $^{210}$Pb, only those undisturbed sites clearly stated in the original literature was included.

Here we did not include unpublished data, as data quality control could be a potential issue. In the peer-reviewed published data, it is assumed that the originating authors and editors have undertaken the necessary steps to verify data quality. All concentrations in air were converted to mBq m$^{-3}$ and all

depositional fluxes to Bq m$^{-2}$ y$^{-1}$, if not already reported in these units. When available, the metadata of latitude, longitude, altitude, sampling date, annual precipitation, methodology, and references are also given. A brief description of different variables that could affect the data is summarized in Table 1. In the event that the air concentration and depositional flux data were only available graphically, a computer



program was used to digitize the data from graphics, the same was done for geographical location and

annual precipitation. In rare cases, only the locality name of the study site was available, the geographical

location was digitized by Google Earth.

**Table 1. Fields in the main data table.**

| Field name | Field description | Field type | Unit |
|---|---|---|---|
| Site | Locality name (city, country) or station name of study site | Short text | Unitless |
| Sampling time | Sampling period/date in month/year format | Short text | Unitless |
| Latitude | North latitude in decimal degrees (from -90 to +90) either directly from original studies or extracted from Google Earth | Number | ° |
| Longitude | East longitude in decimal degrees (from -180 to +180) either directly from original studies or extracted from Google Earth | Number | ° |
| Altitude | Altitudes of study sites either directly from original studies | Number | m |
| Annual precipitation | Mean annual precipitation of study sites directly from original studies | Number | mm |
| Sampling device[a] | Model of aerosol sampling device and air flow rate during sampling | Short text | Unitless |
| Filter[a] | Model, material and dimension of filter membranes | Short text | Unitless |
| Frequency[a] | Sampling frequency set during observation | Short text | Unitless |
| Data number[a] | Total number of measurements performed | Integer | Unitless |
| Concentration[a] | $^{7}$Be or $^{210}$Pb annual concentration in surface air | Number | $mBq\ m^{-3}$ |
| Con-error[a] | Uncertainty in $^{7}$Be or $^{210}$Pb annual concentration in surface air | Number | $mBq\ m^{-3}$ |
| Con-range[a] | Minimum to maximum values of $^{7}$Be or $^{210}$Pb concentration | Number | $mBq\ m^{-3}$ |
| Method[b] | Short descriptor of $^{7}$Be or $^{210}$Pb depositional flux obtained from different methods | Short text | Unitless |
| Rain collector[b] | Material, shape and collection area of rain collectors | Short text | Unitless |
| Clean procedure[b] | Rainwater collector cleaning process and reagents used | Short text | Unitless |
| Preconcentration[b] | Preconcentration method for rainwater samples | Short text | Unitless |
| Recovery[b] | Recovery of $^{7}$Be or $^{210}$Pb and the determination method | Short text | Unitless |
| Decay-correction[b] | Calculation of decay-correction for $^{7}$Be or $^{210}$Pb | Short text | Unitless |
| Flux[b] | $^{7}$Be or $^{210}$Pb annual depositional flux derived from different methods | Number | $Bq\ m^{-2}\ y^{-1}$ |
| Flux-error[b] | Uncertainty in $^{7}$Be or $^{210}$Pb annual depositional flux | Number | $Bq\ m^{-2}\ y^{-1}$ |
| Contribution of dry deposition[b] | Fraction of dry depositional flux to total depositional flux of $^{7}$Be or $^{210}$Pb | Number | Unitless |
| Reference | Investigator and published year of references | Short text | Unitless |
| Reference's DOI | Digital Object Identifier of references | Short text | Unitless |

[a]The field only appears in $^{7}$Be or $^{210}$Pb annual concentration worksheet;

[b]The field only appears in $^{7}$Be or $^{210}$Pb annual depsositional flux worksheet.



## 3 Results and discussions

### 3.1 Scope of the dataset

From 456 references (Appendix A), we have compiled a comprehensive dataset of atmospheric [7]Be and [210]Pb measurements made by numerous laboratories. The dataset includes 494 annual surface air concentration data of [7]Be covering 367 different sites, 366 annual surface air concentration data of [210]Pb from 270 different sites, 304 annual depositional flux data of [7]Be from 279 different sites, and 645 annual depositional flux data of [210]Pb from 602 different sites. In some cases, data collected from different periods were published in different literature. In these cases, all data from the same site are listed separately in the dataset, but in the data analysis and plotting of this article, the data from the same site are merged. The sampling locations of this global dataset show broad geographical coverage (Fig. 1). The sampling maps are mainly composed of continental sites, while the oceanic monitoring sites are relatively few.

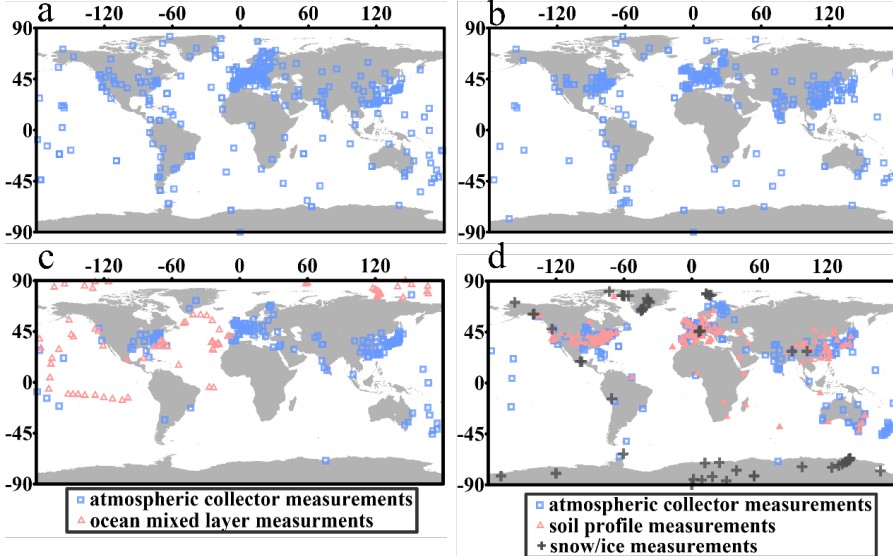

**Figure 1: Maps showing the distribution of sampling locations of (a) [7]Be concentration in surface air, (b) [210]Pb concentration in surface air, (c) [7]Be atmospheric depositional flux, and (d) [210]Pb atmospheric depositional flux.**

A number of peer-reviewed journal articles published annually, containing [7]Be and [210]Pb data from 1955 to 2020 is plotted in Fig. 2. The [7]Be measurements began in the mid-1950s (Arnold and Al-Salih, 1955; Cruikshank et al., 1956; Rama and Zutshi, 1958), earlier than that of [210]Pb, which began in the early



1960s (Burton and Stewart, 1960; Crozaz et al., 1964; Peirson et al., 1966). The long-term monitoring work was started in the 1980s with the $^7$Be and $^{210}$Pb concentration data generated by the EML Surface

Air Sampling Program (Feely et al., 1989; Larsen et al., 1995). This was followed by more ambitious international programs such as the REM network (Hernandez-Ceballos et al., 2015) and IMS-CTBTO (Terzi and Kalinowski, 2017). However, in these two programs, only a few stations measured $^{210}$Pb concentration (Heinrich et al., 2007; Sangiorgi et al., 2019). In contrast, direct $^7$Be and $^{210}$Pb flux measurements were rarely supported by the international program, but there were several national

monitoring programs initiated by developed countries like Australia (Bonnyman and Molina-Ramos, 1976), Japan (Narazaki et al., 2003; Yamamoto et al., 2006), United States (Lamborg et al., 2013) and Finland (Paatero et al., 2015; Leppanen, 2019). The measurement of $^7$Be inventory in ocean mixed layer began in the 1970s (Silker, 1972; Young and Silker, 1974, 1980), and the idea proposed by Young and Silker was subsequently developed in the upper 100-200 m to assess surface water subduction, oxygen

utilization and rate of upwelling (Kadko, 2009; Kadko and Olson, 1996; Kadko and Johns, 2011). The measurement of $^{210}$Pb$_{ex}$ in the undisturbed soil profile was first conducted by Fisenne (1968). Subsequently, Benninger et al. (1975) and Moore and Poet (1976) showed that excess $^{210}$Pb activities in undisturbed soil profiles can be utilized to estimate the atmospheric $^{210}$Pb depositional flux, which motivated an increase in the measurements of $^{210}$Pb$_{ex}$ in soil profiles in the late 1980s (Graustein and

Turekian, 1986, 1989; Monaghan et al., 1989; Dörr and Munnich, 1991). A second boost to soil $^{210}$Pb measurements occurred after the development of $^{210}$Pb$_{ex}$ tracing soil erosion studies on agricultural land (Walling and He, 1999; Walling et al, 2003).

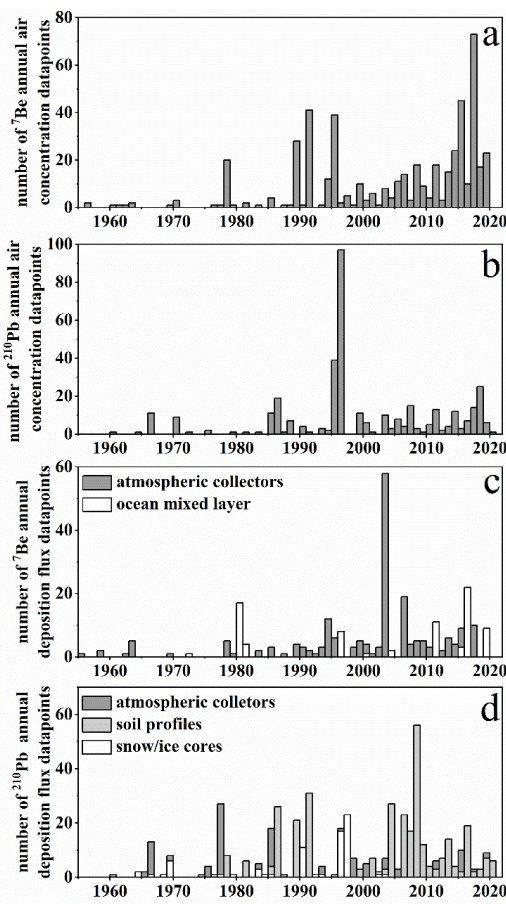

**Figure 2: Time distributions of data published between 1955 and early 2020: (a) [7]Be concentration, (b)[210]Pb**

**concentration, (c) [7]Be depositional flux, and (d) [210]Pb depositional flux.**

The histogram of sampling durations of [7]Be and [210]Pb measurements is given in Fig. 3. In general, the

duration of sampling for [7]Be measurements, especially for air concentration, is longer than that of [210]Pb.

Globally, there are 140 sites that monitored [7]Be air concentration for more than 10 years. The long-term

(decades) measurements of [7]Be were mainly to investigate the negative effect of changes in sunspot

number on [7]Be (Megumi et al., 2000; Cannizzaro et al., 2004; Kulan et al., 2006; Pham et al., 2013;

Steinmann et al., 2013). Due to the simpler measurement procedure for air concentration, the duration of

air concentration measurements, whether for [7]Be or [210]Pb, is generally longer than that of [210]Pb and/or

[7]Be depositional flux measurements.

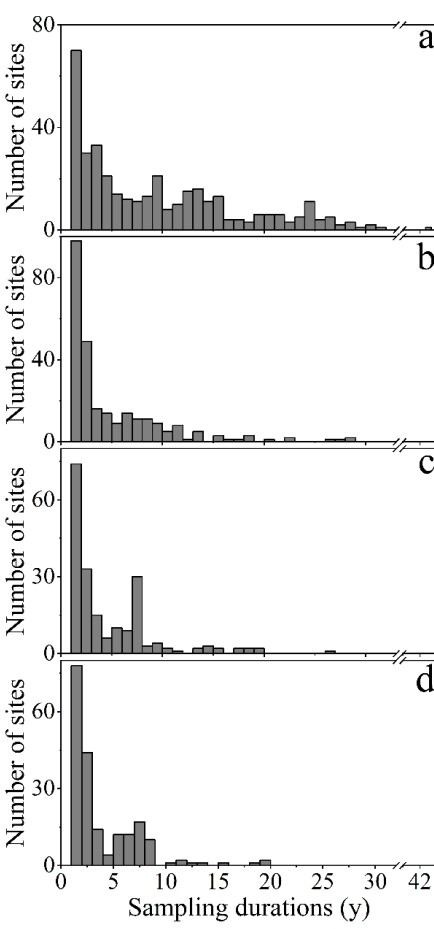

**Figure 3: Histogram of sampling durations of (a) [7]Be concentration, (b) [210]Pb concentration, (c) [7]Be depositional flux, and (d) [210]Pb depositional flux. For those sites with multiple data, we added the sampling duration together, after deducting the overlapping period, if any. Note that the data of [7]Be and [210]Pb depositional flux plotted here only refer to that obtained by using rain collectors.**

**3.2 Global variability**

The global data of [7]Be and [210]Pb air concentrations and depositional fluxes are presented in Fig. 4. The concentration and depositional flux ranges of [7]Be and [210]Pb are 0.33-17.77 mBq m$^{-3}$ and 0.003-4.65 mBq m$^{-3}$, 59-6350 Bq m$^{-2}$ y$^{-1}$ and 1-2539 Bq m$^{-2}$ y$^{-1}$, respectively. The concentrations and depositional fluxes of [7]Be show clear latitudinal variability (Fig. 5a and 5c). In general, [7]Be concentration and flux peak at the mid-latitudes and decrease toward the equator and poles, as was theoretically predicted by Lal and

Peters (1967). A symmetric pattern is observed between the Northern and Southern hemispheres; however, a sharp increment in [7]Be air concentration (lack of flux data) occurred on the Antarctic continent.

Although the $^{210}$Pb concentration and depositional flux are expected to heavily depend on the source(s) of air mass(es), and not to depend on the latitude, the 10° latitudinal variability of $^{210}$Pb concentration and depositional flux in the Northern and Southern hemisphere is observed (Fig. 5b and 5d). The

latitudinal variability of $^{210}$Pb flux is similar to the global fallout curve based on 167 global sites (Baskaran, 2011). Since most of the $^{210}$Pb data are derived from continental sites, the latitudinal variation is mostly due to differences in the radon emanation rates with latitude. As the area of the landmass in the Southern hemisphere is smaller compared to the Northern hemisphere, the $^{210}$Pb concentration and depositional flux are much lower there. An asymmetry is observed in the $^{210}$Pb concentration and

depositional flux between the Northern and Southern hemispheres, with the highest values appearing in the mid-latitudes of the northern hemisphere.

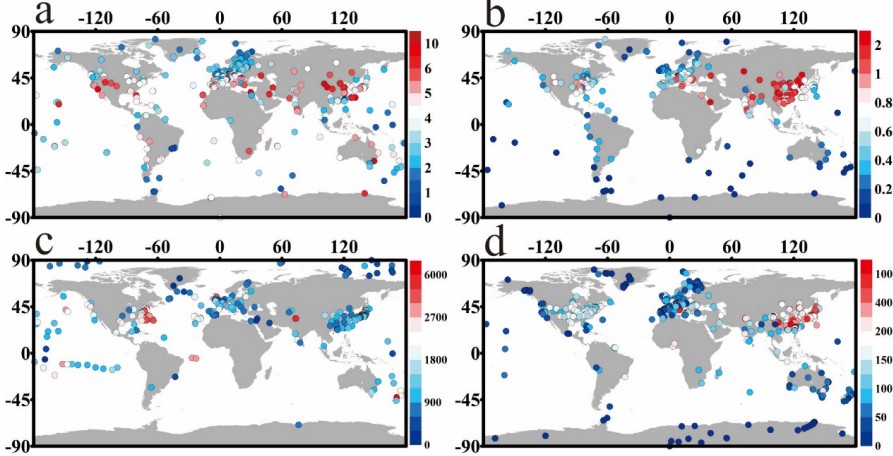

**Figure 4: Global distribution of (a) $^7$Be annual concentration (in mBq m$^{-3}$), (b) $^{210}$Pb annual concentration (in mBq m$^{-3}$), (c) $^7$Be annual depositional flux (in Bq m$^{-2}$ y$^{-1}$) and (d) $^{210}$Pb annual depositional flux (in Bq m$^{-2}$ y$^{-1}$).**

**For those sites with multiple data published, a weighted average was performed based on the sampling durations.**

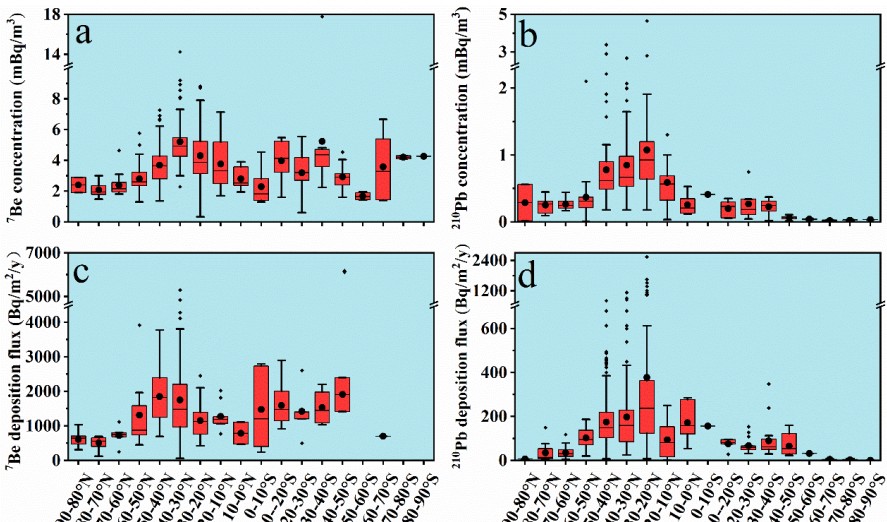

**Figure 5: Latitudinal variability (box whisker plots) of (a) [7]Be concentration, (b) [210]Pb concentration, (c) [7]Be depositional flux, and (d) [210]Pb depositional flux sorted by the 10° latitudinal band.**

**3.3 Contribution of dry deposition and effect of precipitation on depositional flux**

It was reported that dry deposition of [7]Be and [210]Pb generally accounts for less than 10% of the total deposition (Talbot and Andren, 1983; Brown et al., 1989; Todd et al., 1989), however, the fraction of dry deposition of [7]Be and [210]Pb is highly variable (McNeary and Baskaran, 2003; Pham et al., 2013). It is likely that the contribution of dry fallout could increase when annual precipitation decreases (McNeary

and Baskaran, 2003). The fraction of dry to total depositional flux of [7]Be and [210]Pb are presented in Fig. 6a and 6b. Globally, the fraction of dry to total depositional flux of [7]Be and [210]Pb ranged from 1% to 44% (mean:12±9%, n=29, excluding one extreme site without precipitation) and from 5% to 51% (mean: 21±12%, n=26), respectively (Fig. 6c). The low fraction of dry to total depositional fluxes of [7]Be and [210]Pb suggest that these nuclides are removed from the atmosphere primarily by precipitation (both rain

and snowfall). Our results also support previous studies (Baskaran et al., 1993; Benitez-Nelson and Buesseler, 1999) that the fraction of dry deposition is higher for [210]Pb than [7]Be. The fraction of dry fallout of [7]Be and [210]Pb is plotted against annual precipitation in Fig. 6d and 6e, however, a weak negative correlation is observed especially for [210]Pb.

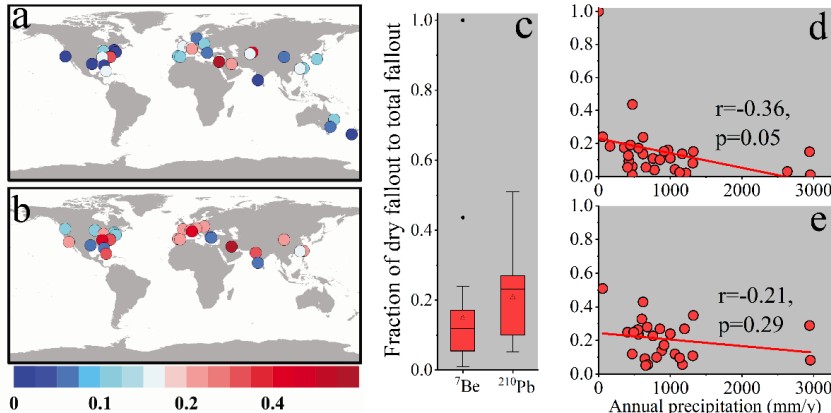

**Figure 6: Global distribution of the fraction of dry to total depositional fluxes of (a)[7]Be and (b) [210]Pb. (c) Box whisker plots showing the comparison between the fraction of dry to total depositional fluxes of [7]Be and the fraction of dry to total depositional fluxes of [210]Pb. (d) The fraction of dry to total depositional fluxes of [7]Be versus annual precipitation. (e) The fraction of dry to total depositional fluxes of [210]Pb against with annual precipitation.**

As precipitation is the primary mechanism of removal of these nuclides from the atmosphere, the annual depositional fluxes of these nuclides generally depend on the amount and frequency of precipitation. In our dataset, the world's lowest [7]Be depositional flux (only 59 Bq m$^{-2}$ y$^{-1}$, less than 5% of that in the same latitude) was observed in a precipitation area, the Judean Desert (Belmaker et al., 2011). The highest [7]Be (6350 Bq m$^{-2}$ y$^{-1}$) and [210]Pb (2539 Bq m$^{-2}$ y$^{-1}$) depositional flux were observed in heavy rainfall areas, Hokitika (Harvey and Matthews, 1989) and Taiwan (Huh and Su, 2004), respectively. Positive correlations between annual depositional flux and precipitation have been observed on a local scale (e.g. Narazaki et al., 2003; Garcia-Orellana et al., 2006; Sanchez-Cabeza et al, 2007; Leppanen et al., 2019). Here we illustrate the effect of precipitation on annual depositional flux on a global scale (Figs. 7 and 8). As both [7]Be and [210]Pb depositional flux show clear latitudinal variability, the fitting curve of annual depositional flux to annual precipitation is plotted within the 10° latitudinal band (if data are available). [7]Be and [210]Pb annual depositional fluxes generally show a good positive correlation with annual precipitation, although the data are limited in some latitudinal bands.


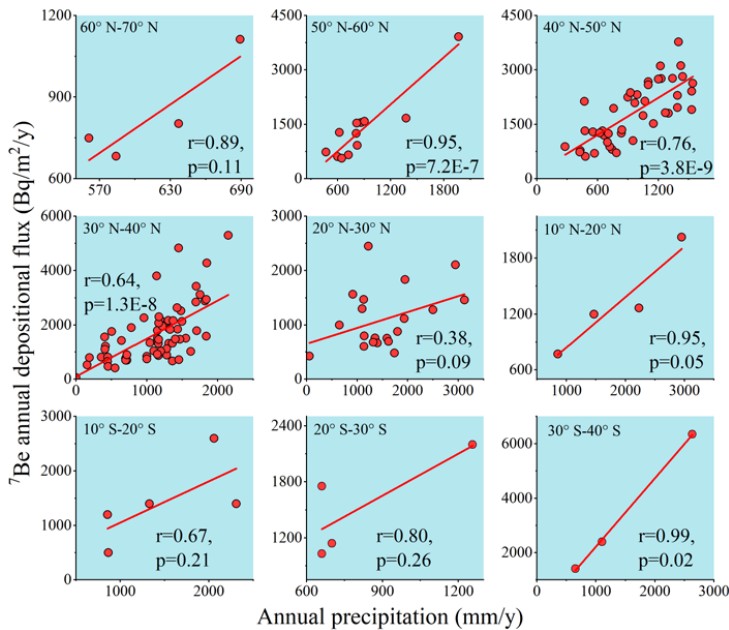

**Figure 7: Annual depositional fluxes of [7]Be are plotted against annual precipitation within the 10° latitudinal band.**

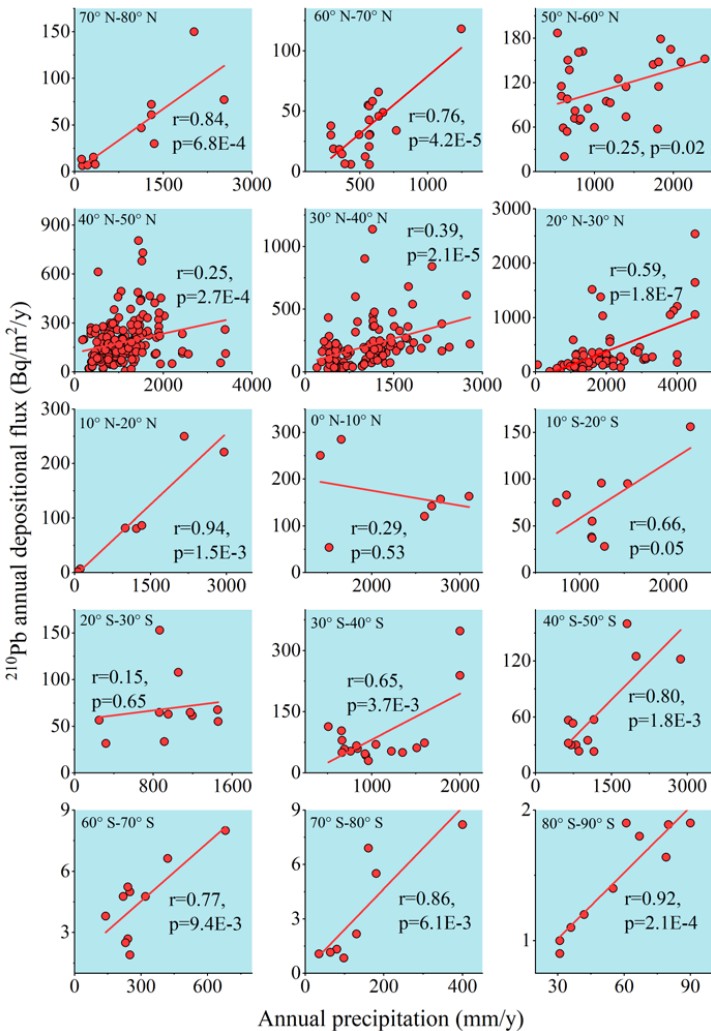

**Figure 8: Annual depositional fluxes of $^{210}$Pb are plotted against annual precipitation within the 10° latitudinal band.**

### 3.4 $^7$Be/$^{210}$Pb ratios and deposition velocities

Considering that some data come from the same station, we further calculated the ratios of $^7$Be to $^{210}$Pb

and deposition velocities of aerosols using $^7$Be and $^{210}$Pb data, as shown in Figs. 9 and 10.

The variations in the $^7$Be/$^{210}$Pb ratios reflect both vertical and horizontal transport in the atmosphere

(Baskaran, 1995; Koch et al., 1996; Arimoto et al., 1999; Lee et al., 2007; Tositti et al., 2014). Our dataset

exhibits similar global patterns of $^7$Be/$^{210}$Pb ratio as simulated with a three-dimensional chemical tracer

model (Koch et al., 1996), with a positive south poleward gradient and a little variation in the northern

hemisphere. Globally, the $^7$Be/$^{210}$Pb air concentration ratio ranged from 2 to 222, and the $^7$Be/$^{210}$Pb

depositional flux ratio ranged from 2 to 229. In 19 sites, $^7$Be/$^{210}$Pb air concentration ratio and depositional

flux ratio are available simultaneously, the paired t-test indicates that at 0.05 level the $^7$Be/$^{210}$Pb air

concentration ratio and depositional flux ratio are not significantly different.

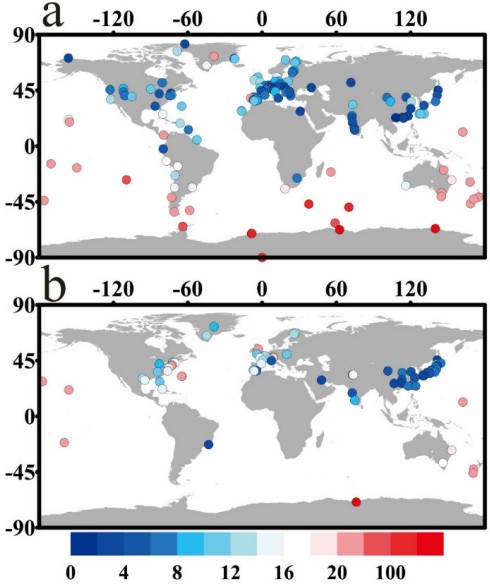


**Figure 9: Global distribution of (a) $^7$Be/$^{210}$Pb activity ratio and (b) $^7$Be/$^{210}$Pb depositional flux ratio. Note that the color bar below applies to both figures.**

$^7$Be and $^{210}$Pb are excellent tracers for the determination of the deposition velocities of aerosols for several

reasons: (1) their measurements are easy; (2) their production rates remain fairly constant over a long

period; and (3) their size distribution in aerosols are similar to that of many particulate contaminants of

interest (McNeary and Baskaran, 2003; Dueñas et al., 2005). When both air concentration (C) and

depositional flux (F) at the same site are available, the average total deposition velocities of aerosols that

carry these nuclides ($V_d$) can be calculated by the following Eq. (1):

$$V_d = F/C ,$$  (1)

Thus, the $V_d$ obtained from $^7$Be and $^{210}$Pb can be used to determine the depositional flux of analog species

(Turekian et al., 1983) with a knowledge of the air concentration of these analog species, assuming that

the scavenging behavior of analog species is similar to $^7$Be and $^{210}$Pb. The $V_d$ for $^7$Be ranged from 0.18-

8.39 (mean: 1.25±1.16, n=70) cm s$^{-1}$, and for $^{210}$Pb ranged from 0.13-12.70 (mean: 1.16±1.68, n=72) cm



s⁻¹. The deposition velocity of aerosols collected over a period of 17 months in Detroit, MI, USA varied

over two orders of magnitude, from 0.2 to 3.6 (mean: 1.6, n=30) cm s⁻¹ for $^7$Be and 0.04 to 3.6 cm s⁻¹

(mean: 1.1, n=30) cm s⁻¹ (McNeary and Baskaran, 2003). A summary of depositional velocity from 10

different stations are also given in McNeary and Baskaran (2003). Earlier studies suggested that, at

continental sites, $V_d$ of $^7$Be will be higher than $V_d$ of $^{210}$Pb using the ground level as the reference, which

is an artifact of the manner in the calculation (Turekian et al., 1983; Todd et al., 1989; McNeary and

Baskaran, 2003). However, later works observed opposite results (Dueñas et al., 2005, 2017; Lozano et

al., 2011; Mohan et al., 2019). The independent t-test analysis indicates that at 0.05 level the $V_d$ calculated

by $^7$Be and $^{210}$Pb are not significantly different in the global dataset, which suggests that $^7$Be and $^{210}$Pb

attach onto the aerosols by similar mechanisms (Winkler et al., 1998; Papastefanou, 2006), and are

affected by similar deposition processes (Lozano et al., 2013).

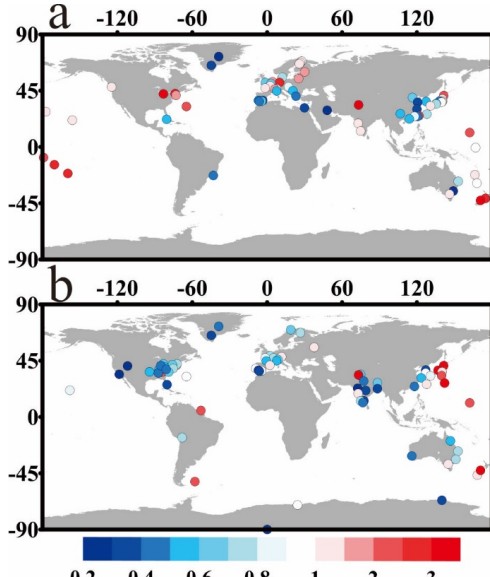

**Figure 10: Global distribution of deposition velocities ($V_d$) of aerosols (in cm s⁻¹) obtained from (a)$^7$Be and (b) $^{210}$Pb. Note that the color bar below applies to both figures.**

**3.5 Investigations on global atmospheric dynamics and climate changes**

Developing numerical models in which aerosols, chemistry, radiation, and clouds interact with one

another and with atmospheric dynamics is important for understanding and predicting global climate

changes (Brost et al., 1991). In such atmospheric dynamic models, the major uncertainty is from the

parameterization of subgrid-scale processes such as precipitation scavenging, vertical transport, and

radiative effect. The cosmogenic $^7$Be and terrigenous $^{210}$Pb, taken together, offer an excellent tool in investigating wet scavenging and vertical transport in global models (Liu et al., 2001).

A set of data obtained prior to the 1990s were used to compare simulated results in global models such as ECHAM2 (Brost et al., 1991; Feichter et al, 1991), ECMWF (Rehfeld and Heimann, 1995), CTM (Balkanski et al., 1993; Koch et al., 1996), GEOS-CHEM (Liu et al., 2001) and LMDz (Preiss and Genthon, 1997; Heinrich and Pilon, 2013). By simulating the ratio $^7$Be/$^{210}$Pb, Koch et al. (1996) eliminated the error associated with the effect of precipitation and provided a better measure of vertical

transport. After correcting the cross-tropopause transport, simulation of observed $^7$Be and $^{210}$Pb surface concentrations and depositional fluxes with no significant global bias was obtained (Liu et al., 2001). Note that the spatial coverage of the dataset used in the previous modeling work was only partial, thus, reducing the statistical significance of comparisons of simulated and observed results (Feither et al., 1991). Additional work with more data is needed for detailed comparison and successful validation of

models (Brost et al., 1991). The size of the $^7$Be and $^{210}$Pb datasets has greatly increased in the last three decades and our new dataset is expected to lay a foundation to develop better parameterization and contribute to modeling efforts.

**3.6 Soil erosion, Particle dynamics and ocean surface process studies**

The atmospheric depositional flux data of $^7$Be and $^{210}$Pb are useful in utilizing these nuclides as tracers

for soil erosion and redistribution studies in terrestrial environments (Mabit et al., 2008) as well as particle dynamics study in aqueous environments (Du et al., 2012). The basic principles involved in using of $^7$Be or $^{210}$Pb$_{ex}$ as soil tracers are the same, which is to compare the measured inventory of $^7$Be or $^{210}$Pb$_{ex}$ (Bq m$^{-2}$) at a sampling point with the inventory of an undisturbed (or reference) site (Blake et al., 1999; Walling and He, 1999). Depletion of the inventory means that soil erosion has occurred, whereas

exceeding provides evidence of accumulation and/or redistribution of surficial soil. The first step in these studies is to select a suitable undisturbed site and obtain the reference inventory (Mabit et al., 2009). However, as human activity intensifies, such undisturbed sites are not always readily available. In the estuarine and coastal areas, the mass balance calculations of $^7$Be and $^{210}$Pb$_{ex}$ have become powerful tools to understand the sediment source, transportation, and scavenging processes (e.g. Feng et al., 1999; Su

and Huh, 2002; Du et al., 2010; Saari et al., 2010; Huang et al., 2013). The measured enhanced sediment inventory of $^{210}$Pb$_{ex}$ compared to atmospheric deposition-based inventory (=annual deposition, Bq m$^{-2}$ y$^-$

[1]/ mean-life of the isotope, y) indicates notable sediment focusing or additional particle input other than atmospheric fallout (Swarzenski et al., 2006; Lepore et al., 2009; Wang et al., 2020). Thus, the atmospheric depositional flux data are also important for tracing particle dynamics using $^7$Be and $^{210}$Pb$_{ex}$.

We believe that the atmospheric depositional flux data presented in our dataset will benefit and facilitate soil or coastal sediment erosion/focusing and particle dynamics studies.

The atmospheric depositional flux (or ocean inventory) data of $^7$Be serve as an indispensable parameter for tracing surface ocean process (e.g. subduction, upwelling, and depositional flux of trace metals) (Kadko, 2017; Kadko and Olsen, 1996; Kadko et al., 2015). Due to the low activity of $^7$Be in open ocean

waters, usually, 400-700 L seawater is needed, which imposes some limitations for sampling, especially for deep layers. This constraint has hampered its application. As mentioned above, $^7$Be depositional flux is independent of longitude and is constant over broad latitudinal bands. Thus, the $^7$Be depositional flux data in our dataset can be used to estimate $^7$Be ocean inventory in the same latitude, which can avoid the collection of the large volume of seawater samples and extend the application of $^7$Be in the Open Ocean.

**3.7 Gaps and recommendations**

Our dataset represents an ambitious expansion in comparison to the $^7$Be or $^{210}$Pb datasets currently available (Brost et al., 1991; Preiss et al., 1996; Persson et al., 2016). Although the spatial coverage of this dataset is more significant, it is still unevenly distributed. Compared to depositional flux, the coverage of air concentration is large. The air concentration measurement at oceanic sites is adequate,

but the depositional flux measurement at oceanic sites is rare. Concerning air concentrations in areas such as Europe, East Asia, eastern Oceania, and the eastern United States are well covered, whereas other areas such as the African continent and Northern Asia are underrepresented. A similar spatial coverage pattern exists for the depositional flux of these nuclides, but the regional gaps are more notable, especially for $^7$Be flux data which almost non-existent in Antarctica and African continents. In addition, it needs to

be emphasized that the number of sampling sites, in which both concentration and flux of $^7$Be and $^{210}$Pb were measured simultaneously, are limited. Finally, we acknowledge that the seasonal information is indeed not much discussed for this dataset.

We recommend that future studies should pay more attention to those areas that are currently undersampled or unsampled to better characterize the expected global variability in the $^7$Be and $^{210}$Pb air

concentrations and depositional fluxes, by measuring both nuclides simultaneously to obtain more data

as well as $^{7}$Be/$^{210}$Pb ratio and estimate deposition velocity of aerosols. In areas with very limited precipitation such as deserts, it is expected that the dry fallout will dominate the bulk depositional flux, and quantification of the role of dry fallout in the removal of these nuclides will provide insights on the removal of other analog species. Besides, the size distribution of aerosols particles carrying $^{7}$Be and $^{210}$Pb

is crucial for atmospheric behavior, and such studies need to be strengthened. Further compilation of monthly data is also warranted to assess seasonal variability of $^{7}$Be and $^{210}$Pb and understand the relationship between these changes and influencing factors such as atmospheric dynamics, meteorological condition, and geographic location on a global scale. As mentioned earlier, combining cosmogenic $^{7}$Be with $^{210}$Pb that has a predominantly Earth-surface origin will be useful to trace species

that originate both from Earth's surface, such as Hg, SO$_4^-$, NO$_3^-$, and those that originate in the upper atmosphere, such as O$_3$. The troposphere contains ~99% of global water vapor with < 1% in the stratosphere. The depositional velocity of aerosol in the stratosphere is very low (~ $4 \times 10^{-3}$ cm s$^{-1}$; Junge, 1963) with no precipitation. Thus, the $^{7}$Be concentration is governed by local production, meridional, and vertical downward transport and its decay. Between the upper tropopause and cloud condensation

height, the removal rate of aerosols is also slow. Collection of air samples about cloud condensation height to tropopause will provide useful information on the settling velocity of aerosols (Lal and Baskaran, 2012).

**4 Data formats and availability**

For clarity and convenience, four separate worksheets, each named as $^{7}$Be or $^{210}$Pb annual air

concentration and $^{7}$Be or $^{210}$Pb annual atmospheric flux, are available in one Microsoft Excel® file, although sometimes these data come from the same literature. The dataset can be downloaded from Zenodo (https://doi.org/10.5281/zenodo.4521649, Zhang et al., 2021). It is free for scientific applications, but the free availability does not constitute a license to reproduce or publish it.

**5 Conclusions**

This paper summarizes the global dataset of $^{7}$Be and $^{210}$Pb for their concentration in atmospheric air and their depositional fluxes from 456 publications spanning the time from 1955 to early 2020. The calculated activity ratios of $^{7}$Be/$^{210}$Pb and deposition velocity of aerosols are also reported. Some noteworthy spatial



gaps in the dataset are the African continent, Northern Asia, and Antarctica (only for [7]Be flux). Despite

these gaps, our dataset is the largest compilation of [7]Be and [210]Pb air concentration and depositional flux

up to date and could be used to better understand the transport processes of air masses and depositional

processes of aerosols. This dataset not only lays a solid foundation to develop better parameterization

contributing to future modeling efforts but also supply a basic parameter for tracing soil erosion, particle

dynamics, and ocean surface process using [7]Be and/or [210]Pb.

**Appendix A: List of references in the dataset**

Aaboe et al. (1981), Aba et al. (2016), Ahmed et al. (2004), Akata et al. (2008, 2015, 2018a, 2018b),

Akram et al. (1999), Al-Azmi et al. (2001), Alegría et al. (2010), Ali et al. (2011a, 2011b), Alonso-

Hernández et al. (2004, 2014), Amano and Kasai (1981), Anand and Rangarajan (1990), Anderson et al.

(1960), Andres (2018), Appleby et al. (2002, 2003, 2019), Arimoto et al. (1999), Arkian et al. (2010),

Arnold and Al-Salih (1955), Azahra et al. (2003, 2004), Azimov et al. (2011, 2017), Bachhuber and Bunzl

(1992), Baeza et al. (1996, 2016), Bas et al. (2017), Baskaran and Swarzenski (2007), Baskaran et al.

(1993), Batraov et al. (2016), Bazarbaev et al. (2012), Begy et al. (2016), Beks et al. (1998), Belmaker

et al. (2011), Belyaev et al. (2004), Benitez-Nelson and Buesseler (1999), Benmansour et al. (2013),

Bettoli et al. (1995), Bikkina et al. (2015), Blake et al. (2009), Blazej and Mietelski (2014), Bleichrodt

(1978), Bleichrodt and van Abkoude (1963), Bourcier et al. (2011), Brandt et al. (2018), Branford et al.

(2004), Brost et al. (1991), Brown et al. (1989), Buck et al. (2019), Buraeva et al. (2013a, 2013b), Burton

and Stewart (1960), Caillet et al. (2001), Cámara-Mor et al. (2011), Cannizzaro et al. (1999, 2004),

Canuel et al. (1990), Cao et al. (2018), Carpenter et al. (1981), Carvalho et al. (1995, 2013), Chae and

Kim (2019), Chae et al. (2011), Chang et al. (2008), Chao et al. (2012, 2014), Chen (2014), Chen et al.

(2016, 2020), Chham et al. (2017, 2018, 2019), Cho et al. (2011), Clifton et al. (1995), Conaway et al.

(2013), Courtier et al. (2017), Crecelius (1981), Crozaz and Langway (1966). Crozaz et al. (1964),

Cruikshank et al. (1956), Cruz et al. (2019), Cui et al. (2012), Daish et al. (2005), Damatto et al. (2005),

Damnati et al. (2013), D'Amours et al. (2013), de Tombeur et al. (2020), Deng et al. (2020), Dibb (1989,

1990a, 1990b, 1992, 2007), Dibb and Jaffrezo (1993), Dibb et al. (1994), Ding et al. (2017), Dlugosz-

Lisiecka (2019), Doering and Akber (2008a, 2008b), Doering and Saey (2014), Doering et al. (2006),

Doi et al. (2007), Dominik et al. (1987, 1989), Dörr and Münnich (1991), Dovhyi et al. (2017), Du et al.



(2008, 2015), Du and Walling (2012), Dueñas et al. (1999, 2004, 2005, 2009, 2011, 2017), Ďurana et al. (1996), Dutkiewicz and Husain (1985), El-Hussein et al. (2001), Elsässer et al. (2011), Eriksson et al. (2004), Fan et al. (2013), Fang et al. (2013), Feely et al. (1989), Filizok and Ugur Gorgun (2019), Filizok et al. (2013), Fogh et al. (1999), Fukuyama et al. (2008), Fuller and Hammond (1983), Gäggeler et al.

(1983, 1995), Gai et al. (2015), García-Orellana et al. (2006), Garimella et al. (2003), Garspar et al. (2013), Gavini et al. (1974), Gerasopoulos et al. (2001), Gonzalez-Gomez et al. (2006), Gordo et al. (2015), Grabowska et al. (2003), Graham et al. (2003), Graustein and Turekian (1986, 1989), Grossi et al. (2016), Gustafson et al. (1961), Halstead et al. (2000), Harvey and Matthews (1989), Hasebe et al. (1981), Hasegawa et al. (2007), Haskell et al. (2015), He and Walling (1997), He et al. (2018), Heikkilä

et al. (2008), Heinrich et al. (2007), Hernández et al. (2005, 2007, 2008), Hernandez-Ceballos et al. (2015), Hicks and Goodman (1977), Hirose et al. (2004), Hötzl et al. (1987), Houali et al. (2019), Hu (2016), Hu and Zhang (2019), Hu et al. (2020), Huang et al. (2019), Huh and Su (2004), Huh et al. (2006), Igarashi et al. (1998), Ioannidou and Paatero (2014), Ioannidou and Papastefanou (2006), Ioannidou et al. (2005, 2019), Irlweck et al. (1997), Isakar et al. (2016), Ishikawa et al. (1995), Itoh and Narazaki

(2017), Itthipoonthanakorn et al. (2019), Iurian et al. (2013), Jankovic et al. (2014), Jasiulionis and Wershofen (2005), Jia and Jia (2014), Jia et al. (2003), Jiang (1999), Joshi (1985), Joshi et al. (1969), Juri Ayub et al. (2009), Kadko (2000), Kadko and Johns (2011), Kadko and Olson (1996), Kadko and Prospero (2011), Kadko and Swart (2004), Kadko et al. (2015, 2016), Kapala et al. (2018), Karwan et al. (2016), Kato et al. (2010), Khan et al. (2008, 2009), Khodadadi et al. (2018), Kikuchi et al. (2009), Kim

et al. (1998, 1999, 2000, 2005), Kitto et al. (2005, 2006), Klaminder et al. (2006), Koide et al. (1977, 1979), Kolb (1970), Kownacka et al. (1990), Krmar et al. (2015), Kulan et al. (2006), Kurata et al. (1986), Laguionie et al. (2014), Lal et al. (1979), Lambert et al. (1990), Lamborg et al. (2000, 2003), Landis et al. (2014), Larsen et al. (1995), Lee et al. (1985, 2002, 2015), Leppanen (2019), Li et al. (2009, 2013, 2017a, 2017b), Likuku (2006a, 2006b), Lin et al. (2014), Lindblom (1969), Liu et al. (2014), Lockhart

Jr et al. (1966), Lozano et al. (2011, 2012, 2013), Lujanienë (2003), Luyanas et al. (1970), Mabit et al. (2009), Maenhaut et al. (1979), Magno et al. (1970), Marx et al. (2005), Mattsson (1970), McNeary and Baskaran (2003), Megumi et al. (2000), Mélières et al. (2003), Men et al. (2016), Meusburger et al. (2016, 2018), Mietelski et al. (2017), Milton et al. (2001), Miralles et al. (2004), Mohan et al. (2018, 2019), Mohery et al. (2014, 2016), Momoshima et al. (2006), Monaghan (1989), Monaghan and Holdsworth

(1990), Monaghan et al. (1986), Moore and Poet (1976), Muramatsu et al. (2008), Narazaki and Fujitaka

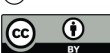



(2009), Narazaki et al. (2003), Neroda et al. (2016), Nijampurkar and Clausen (1990), Nijampurkar and Rao (1993), Nijampurkar et al. (2002), Noithong et al. (2019), Nozaki et al. (1978), O'Farrell et al. (2007), Olsen et al. (1985), Othman et al. (1998), Paatero and Hatakka (2000), Paatero et al. (2003, 2010, 2015, 2017), Pacini et al. (2011, 2015), Padilla et al. (2019), Pan et al. (2011, 2017), Papastefanou and Bondietti

(1991), Papastefanou and Ioannidou (1991), Papastefanou et al. (1995), Parker (1962), Peirson (1963), Peirson et al. (1966), Peng et al. (2019), Perreault et al. (2017), Persson (2016), Peters et al. (1997), Pfahler et al. (2004), Pham et al. (2011, 2013), Picciotto et al. (1964, 1968), Piñero-García and Ferro-García (2013), Piñero-García et al. (2012, 2015), Poet,et al. (1972), Poreba et al. (2019), Porto and Walling (2012), Porto et al. (2006, 2009, 2013, 2014, 2016), Pourchet et al. (1997), Preiss et al. (1996),

Prospero et al. (1995), Qian et al. (1985), Rabesiranana et al. (2016), Rajačić et al. (2015, 2016), Raksawong et al. (2017), Ram and Sarin (2012), Rangarajan et al. (1966, 1975, 1986), Rastogi and Sarin (2008), Realo et al. (2004, 2007), Reiter et al. (1983), Renfro et al. (2013), Rodas Ceballos et al. (2016), Ródenas et al. (1997), Rodriguez-Perulero et al. (2019), Saari et al. (2010), Sabuti and Mohamed (2016), Sakurai et al. (2005, 2011), Saleh and Abdel-Halim (2017), Sambayev et al. (2019), Samolov et al. (2014),

San Miguel et al. (2019), Sanchez-Cabeza et al. (2007), Sanders et al. (2011), Sato et al. (1994, 2003), Savva et al. (2018), Schuler et al. (1991), Schumann and Stoeppler (1963), Shapiro and Forbes-Resha (1976), Sheets et al. (1999), Shelley et al. (2016), Shi et al. (2011, 2017), Shleien and Friend (1966), Short et al. (2007), Silker (1972), Simon et al. (2009), Smith et al. (1997), Song et al. (2003, 2015), Stamoulis et al. (2018), Steinmann et al. (1999, 2013), Stromsoe et al. (2016), Su et al. (2003), Sugihara

et al. (2000), Suzuki and Shiono (1995), Suzuki et al. (1999, 2004, 2017), Sykora et al. (2017), Talbot and Andren (1983), Tan et al. (2013, 2016), Tanahara et al. (2014), Tanaka and Turekian (1995), Tateda and Iwao (2008), Taylor et al. (2016), Terzi and Kalinowski (2017), Thang et al. (2018), Thompson et al. (1984), Thor and Zutshi (1958), Todd et al. (1989), Todorovic et al. (1999, 2000, 2005, 2010), Tokieda et al. (1996), Tositti et al. (2014), Tsunogai,et al. (1985, 1988), Tuo et al. (2018), Turekian and Cochran

(1981), Turekian et al. (1977, 1983), Uchida et al. (2009), Uematsu et al. (1994), Ueno et al. (2003), Uğur et al. (2011), Uhlář et al. (2014), Valles et al. (2009), Van Metre and Fuller (2009), Vecchi and Valli (1997), Vecchi et al. (2005), Vogler et al. (1996), Von Gunten et al. (1993), Wagenbach et al. (1988), Wakiyama et al. (2010), Wallbrink and Murray (1994, 1996), Walling and He (1999), Walling et al. (2003), Walton and Fried (1962), Wan et al. (2010), Wang (2010, 2011), Wang et al. (2014a, 2014b),

Weiss and Naidu (1986), Wells et al. (2012), Windom (1969), Winkler and Rosner (2000), Winkler et al.

(1998), Wu et al. (2011), Yamagata et al. (2019), Yamamoto et al. (2006), Yang et al. (1999, 2011, 2013), Yi et al. (2005, 2007), Yoshimori et al. (2005), Young and Silker (1980), Yu et al. (2017, 2018), Zanis et al. (1999, 2003), Zhang et al. (2003, 2006, 2014, 2016, 2018a, 2018b, 2019), Zheng et al. (2005, 2007), Zhu and Olsen (2009).

**Author contributions**

FZ is responsible for most of the writing of this article, along with the assembly of the data and preparation of the figures. QZ and YW assisted FZ in compiling the data. JW, MB, QZ, PJ and JD contributed to the review of the manuscript.

**Competing interests**

The authors declare that they have no conflict of interest.

**Acknowledgements**

We thank all scientists conducting research on $^{7}$Be and/or $^{210}$Pb, whose previous work and published data made our compilation possible. Fule Zhang would like to thank Yufeng Chen for his assistance in preparing graphics.

**Financial support**

This research has been supported by the Science and Technology Plan Projects of Guangxi Province (2017AB30024).

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
