# Peer review of "A global dataset of atmospheric 7Be and 210Pb measurements: annual air concentration and depositional flux"

_Earth System Science Data, 2021_

## Author Comment (AC1)

This paper presents a global data set of surface air concentrations and depositional fluxes of $^7$Be and $^{210}$Pb that the authors compiled from literatures published during 1955-2020. This effort is timely as it has been a long time since last time such a data set was compiled. The two radionuclides are very useful tracers for studying Earth's surface (land/ocean) processes as well as transport and deposition processes in the atmosphere. The new data set is expected to be widely used and cited in the years to come. The content of this paper is generally well presented, but I do have some concerns that should be addressed before its publication on ESSD.

We would like to thank the anonymous referee #2 for taking the time to provide a thorough review of our submitted manuscript. The comments are very valuable and the suggestions are very helpful. These comments and suggestions help us in greatly improving the quality of our MS.
Below, the original comments are in black, our responses are in blue.

Major comments:

(1). There are many typos and grammatical errors in the text. Some are listed below. Editing assistance is needed (perhaps from coauthor MB) and would significantly improve the presentation.
Response: Thank you very much for pointing out the typos and grammatical errors in the manuscript. We have corrected these typos and grammatical errors. Language has been carefully further edited by one of the coauthors MB. Besides, the editorial team of the ESSD will also edit the language if the manuscript is accepted, as presented in the submission guidelines in the homepage of the journal.

(2). "Finally, we acknowledge that the seasonal information is indeed not much discussed for this dataset. (P22, L456-457)"; "Further compilation of monthly data is also warranted to assess seasonal variability of 7Be and 210Pb and understand the relationship between these changes and influencing factors such as atmospheric dynamics, meteorological condition, and geographic location on a global scale. (P23, L465-468)"
---- As authors mentioned in the paper, seasonal air concentrations and depositional fluxes of 7Be and 210Pb are not reported. Such data would otherwise significantly increase the value of this new compilation. For example, the seasonal data can be used to evaluate seasonality of transport in global atmospheric models. The authors are strongly encouraged to add the seasonal data into their data set, if at all possible. If not, a discussion of why the seasonal data are not included would be helpful. In that case, compiling the seasonal data in a future effort is also encouraged.
Response:
We totally agree that seasonal data would significantly increase the value of this new dataset. Actually, seasonal $^7$Be and $^{210}$Pb data has never been compiled on a global scale. We did try to compile some seasonal data of $^7$Be and $^{210}$Pb, but this completion work is incomplete. Most of the data for seasonal studies are presented in graphs not in tables, and in many older papers, the quality of graph and the paper used is poor and have to compromise the precision in extracting the data. Second, in some papers, although seasonal data were measured, only the annual data were provided. Furthermore, wherever there are seasonal data, it is important to have data on the amount of precipitation along with radionuclide data, as seasonal variations on the amount of precipitation plays a major role on the atmospheric scavenging and their depositional flux. Last and most importantly, since we were unable to retrieve reliable data from the graphs/charts, we reached out to some of the original authors for their original data, but received little help. And many of the older references, the authors no more active with their research and/or have retired or no more alive. Due to these constraints, we currently only have compiled only partial seasonal data, which is far from our ultimate goal. Many funding agencies now require that researchers submit their data to a public domain (such as National Science Foundation in USA, GEOTRACES Program) which will be accessible to global scientific community. More funding agencies should encourage to either join such efforts or start one in their home country and such data must be available for global scientific community, with no strings attached. We plan to reach out researchers who have still access to their seasonal data and try our best to compile the seasonal data in a future effort (may be need 1-2 y), then update the current version of the dataset.

In addition to the constraints listed above, adding seasonal data and related discussions will likely make this paper too lengthy, and thus have focused on annual data in the current work. To alleviate the concerns of the reviewer, we have added a short paragraph (given below) at the end of section 3.7 giving the rationale why we have not included seasonal data.

"Finally, we acknowledge that the seasonal data of $^7$Be and $^{210}$Pb has not been included in the current version of dataset, because compiling the seasonal data is more challenging than compiling the annual data. Unlike the annual data, most of the published seasonal data are presented in graphs, without giving in tables, and in some cases, the graph quality was poor and precision in data extraction is expected to be poor. Besides, in some papers, although seasonal data were measured, only the annual data were provided. Thus, the comprehensive compilation of seasonal data of $^7$Be and $^{210}$Pb may need collaboration with and data sharing from the scientific community. The compilation of seasonal data is expected to be useful to assess seasonal variability of $^7$Be and $^{210}$Pb and understand the relationship between these changes and influencing factors such as atmospheric dynamics, meteorological conditions, and geographic location on a global scale. And the seasonal data can also be useful in evaluating seasonality of transport in global atmospheric models."

Because there is no discussion on seasonal variations, the title of this paper is now changed to "A global dataset of atmospheric $^7$Be and $^{210}$Pb measurements: **annual** air concentration and depositional flux"

(3). Is the unit of air concentration "mBq m^-3" or "mBq / SCM" where "SCM" stands for standard cubic meter?

Response: The unit of air concentration is "mBq m$^{-3}$".

(4). P2, L34-35:  "Depositional flux of 7Be is independent of longitude but depends on the altitude and the ~11 years solar cyle"

As Figure 4c shows, the 7Be depositional flux does depend on longitude, and the error bars show the longitudinal variability of 7Be deposition fluxes is quite large at northern mid-latitudes.   Do you mean the production rate of 7Be is independent of longitude? Do you mean "latitude" by "altitude" here?

Response: Thank you for noting the mistake here. Here we originally intended to express that the production rate of [7]Be is independent of longitude. And the word "latitude" was missed here. This sentence is now rewritten as "**The production rate of [7]Be** has negligible dependence on longitude or season, but depends on altitude, latitude and the ~11 years solar cycle (Koch et al., 1996; Liu et al., 2001; Su et al., 2003)". And to make the text more coherent, this sentence will be moved forward at the end of the sentence "[7]Be, a cosmogenic radionuclide, is produced by the spallation of oxygen and nitrogen nuclei by cosmic rays in the stratosphere and upper troposphere."

Reference:

Koch, D. M., Jacob, D. J., and Graustein, W. C.: Vertical transport of tropospheric aerosols as indicated by and in a chemical tracer model, J. Geophys. Res., 101, 18651-18618, 1996.

Liu, H., Jacob, D. J., Hey, I., and Yantosca, R. M.: Constraints from [210]Pb and [7]Be on wet deposition and transport in a global three-dimensional chemical tracer model driven by assimilated meteorological fields, J. Geophys. Res., 106, 12109-12128, 2001.

Su, C. C., Huh, C. A., and Lin, F. J.: Factors controlling atmospheric fluxes of [7]Be and [210]Pb in northern Taiwan, Geophys. Res. Lett., 30, https://doi.org/10.1029/2003GL018221, 2003.

P22, L441-444: "As mentioned above, 7Be depositional flux is independent of longitude and is constant over latitudinal bands. Thus, the 7Be depositional flux data in our dataset can be used to estimate 7Be ocean inventory in the same latitude, which can avoid the collection of the large volume of seawater samples and extend the application of 7Be in the Open Ocean."

Again, see the comment above. In that case, the 7Be depositional flux data in the dataset would not be able to be used to estimate 7Be ocean inventory in the same latitude.

Response:

Indeed, in Figure 4c, the [7]Be depositional flux varies with longitude even within the specific latitudinal bands, but we believe **such variability is mainly due to spatial variations in the amount of precipitation** since [7]Be is removed from atmosphere primarily by precipitation. The dataset supports this observation. As shown in Fig. 7 (see below), **[7]Be annual depositional fluxes generally show a significant positive correlation with annual amount precipitation**, especially at the northern mid-latitudes where the data coverage is good. In this case, the empirical equation between [7]Be depositional fluxes and annual precipitation provide an empirical method for estimating fluxes, although frequency of precipitation also is likely a factor.

To alleviate the concerns of this reviewer and the other reviewer, we have added a new paragraph (as below) outlining and clarifying the use of the dataset. Besides, the empirical equations describing the relationships between annual precipitation and [7]Be depositional fluxes for different latitudinal belts is also added as a new table in the revised manuscript.
"Our dataset provides a forum in which a large amount of $^7$Be and $^{210}$Pb atmospheric
depositional flux data for the above-mentioned research communities. This database
will help in identifying data gaps and evaluating the empirical relations between $^7$Be
and $^{210}$Pb depositional fluxes and annual precipitation. Researchers can rely on
previously collected data in planning their research, without additional monitoring of
$^7$Be and/or $^{210}$Pb depositional fluxes. Even for those areas with data gaps, the empirical
equations between $^7$Be and $^{210}$Pb depositional fluxes and annual precipitation provide
an empirical method for estimating fluxes, especially for $^7$Be, as $^7$Be depositional flux
is independent of longitude and is constant over broad latitudinal bands. In summary,
the atmospheric depositional flux data presented in our dataset along with the meta-
analysis of the data will be useful in the investigations of soil erosion studies in
terrestrial environments, particle dynamics studies in aquatic systems, and surface
mixing process studies in open ocean."

[Figure]

Minor comments:
P1, L29:   Earth's surface AND ATMOSPHERIC processes
Response: Thank you for the suggestion. "and atmospheric" will be added here in the
revised manuscript.
P2, L32-34:   correct grammar.
Response: Thank you for noting this mistake. This sentence is rephrased as "A major
fraction of $^7$Be (67%) production takes place in the stratosphere, but it does not readily
reach the troposphere except during spring when seasonal thinning of tropopause folds near the jet stream take occurs at mid-latitudes (Lal and Peters, 1967; Danielsen, 1968).
Thus, $^7$Be flux to the Earth' surface varies with latitude and season (Lal and Peters,
1967; Koch and Mann, 1996)."

Reference
Danielsen, E. F.: Stratospheric-tropospheric exchange based on radioactivity, ozone, and potential
vorticity. J. Atmos. Sci., 25, 502-518, 1968.
Koch, D. M. and Mann, M. E.: Spatial and temporal variability of $^7$Be surface concentration, Tellus
B, 48, 387-396, 1996.
Lal, D. and Peters, B.: Cosmic ray produced radioactivity on the Earth, in: Handbuch der Physik /
Encyclopedia of Physics, edited by: Sittle, K., Springer, Berlin, Heidelberg, Germany, 551-612,
https://doi.org/10.1007/978-3-642-46079-1_7, 1967.

P2, L56:   studyING
Response: Thank you for noting this mistake – it is corrected in the revised manuscript.

P2, L56:   add comma after all "e.g." throughout the text
Response: Thank you for noting this mistake – we have added comma after all "e.g."
in the revised manuscript.

P3, L80:   fluxes OF 7Be
Response: Thank you for noting this mistake – it is corrected in the revised manuscript.

P3, L84:   "To date, only one dataset was published that compiled 7Be and 210Pb
together (Persson, 2016)" ---   is it actually a 2015 publication?
Persson, B. R. R. (2015) Global distribution of     7Be, 210Pb and, 210Po in the surface
air. Acta Scientiarum Lundensia, Vol.2015-008, pp.1-24.   ISSN 1651-5013
Response: Thank you for noting this mistake – it is corrected in the revised manuscript.
The corresponding reference in the reference list is also corrected.

P4, L111: This is confusing. Correct grammar. Complementary is an adjective.
Response: Thank you for noting this mistake. This sentence is now rephrased as "using
natural archives avoids the labor and time-intensive measurements of $^7$Be and $^{210}$Pb
concentration in precipitation and can serve as a complement to …"

P5, L133:   Alternately -   do you actually mean "Alternatively, "
Response: Thank you for noting this mistake – it is corrected in the revised manuscript.

P6, L158-162:   This sentence is way too long and hard to understand.   Please revise.
Response: Thank you for the suggestion. This sentence is now split into two sentences:
"It is expected that the $^7$Be inventory is season-dependent in areas with large seasonal
variations in precipitation (e.g., monsoon-dominated continental and oceanic areas).
Time-series study in Bermuda has shown that the inventory of $^7$Be was relatively constant throughout the year, such that [7]Be inventory measured at any one time is likely representative (to within 20%) of the instantaneous [7]Be flux (Kadko et al., 2015)."
Reference
Kadko, D., Landing, W. M., and Shelley, R. U.: A novel tracer technique to quantify the atmospheric flux of trace elements to remote ocean regions, J. Geophys. Res-Oceans, 120, 848-858, 2015.

P6, L171:  "and hence those are data are not included" – please rewrite.
Response: Thank you for the suggestion. This sentence is rewritten as "the data of [7]Be soil inventory are not included in our dataset".

P7, L175:  , AND the latter
Response: Thank you for noting this mistake – it is corrected in the revised manuscript.

P7, L200:  remove "the" before "dating ice core"
Response: Thank you for noting this mistake – it is removed in the revised manuscript.

P7, L204:   typo "filed" (field)
Response: Thank you for noting this mistake –it is corrected in the revised manuscript.

P8, L209:  can ALSO be obtained
Response: Thank you for noting this mistake –it is corrected in the revised manuscript. A similar mistake is also corrected.

P8, L223: only those sites WITH more than one year of data
Response: Thank you for noting this mistake – it is corrected in the revised manuscript.

P10, L255:   THE number of
Response: Thank you for noting this mistake –it is corrected in the revised manuscript.

P13, L301:  "a sharp increment in 7Be air concentration occurred on the Antarctic continent" – this reflects the subsiding motion of air over the Antarctic continent
Response: Thank you for the suggestion. This sentence will be rewritten as "a sharp increment in [7]Be air concentration (lack of flux data) occurred on the Antarctic, which reflects the subsidence of stratospheric air masses over the Antarctica continent (Wagenbach et al., 1988; Elsässer et al., 2011)."
Reference
Elsässer, C., Wagenbach, D., Weller, R., Auer, M., Wallner, A., and Christl, M.: Continuous 25-yr aerosol records at coastal Antarctica, Tellus B, 63, 920-934, 2011.
Wagenbach, D., Görlach, U., Moser, K., and Münnich, K. O.: Coastal Antarctic aerosol: the seasonal pattern of its chemical composition and radionuclide content, Tellus, 40B, 426-436, 1988.

P15, Fig.5:  the convention is to plot from South to North (x-axis).  Also indicate what the whiskers / dots / bars stand for.
Response: The Fig. 5 has been replotted as below, and have added a legend to indicate what the whiskers / dots / bars stand for.

[Figure]

P21, L406: "CTM" is the abbreviation for chemical transport model; it's not a model
name.
How about "a CTM based on GISS GCM"?
Response: Thank you for the suggestion. "CTM" will be changed as "CTM based on
GISS GCM". In addition, the model "GMI CTM" (Liu et al., 2016) is also added in this
sentence.
Reference
Liu, H., Considine, D. B., Horowitz, L. W., Crawford, J. H., Rodriguez, J. M., Strahan, S. E., Damon,
M. R., Steenrod, S. D., Xu, X., Kouatchou, J., Carouge, C., and Yantosca, R. M.: Using
beryllium-7 to assess cross-tropopause transport in global models, Atmos. Chem. Phys., 16,
4641-4659, 2016.

P21, L431-432:    Not sure what "Bq m-2 y-1 / mean-life of the isotope, y)" means.
Response: Based on the suggestion of anonymous referee #1, we have made major
revisions (as below) of section 3.6, and this sentence has been deleted in the revised
manuscript.
The sentence in L428-436 is rewritten as "In aquatic systems (including river, lake,
estuary and coast), the mass balance models of $^7$Be and $^{210}$Pb$_{ex}$ have become powerful
tools to understand the sediment source, transportation and resuspension processes (e.g.
Wieland et al., 1991; Feng et al., 1999; Jweda et al., 2008; Huang et al., 2013; Mudbidre
et al., 2014). In such models, the atmospheric depositional input of $^7$Be and $^{210}$Pb is a
required source term. In addition, $^7$Be/$^{210}$Pb$_{ex}$ activity ratio can be used to identify the
source area of sediments (Whiting et al., 2005; Jweda et al., 2008; Wang et al., 2021),
to quantify the age of sediments (Matisoff et al., 2005; Saari et al., 2010), and to
determine the transport distance of suspended particles (Bonniwell et al., 1999,
Matisoff et al., 2002). Thus, the atmospheric depositional flux data of $^7$Be and $^{210}$Pb are
also important for tracing particle dynamics in aquatic systems'

Reference

Bonniwell, E. C., Matisoff, G., and Whiting, P. J.: Determining the times and distances of particle transit in a mountain stream using fallout radionuclides, Geomorphology, 27, 75-92, 1999.

Feng, H., Cochran, J. K., and Hirschberg, D. J.: $^{234}$Th and $^{7}$Be as tracers for the transport and dynamics of suspended particles in a partially mixed estuary, Geochim. Cosmochim. Ac., 63, 2487-2505, 1999.

Huang, D., Du, J., Moore, W. S., and Zhang, J.: Particle dynamics of the Changjiang Estuary and adjacent coastal region determined by natural particle-reactive radionuclides ($^{7}$Be, $^{210}$Pb, and $^{234}$Th), J. Geophys. Res-Oceans, 118, 1736-1748, 2013.

Jweda, J., Baskaran, M., van Hees, E., and Schweitzer, L.: Short-lived radionuclides ($^{7}$Be and $^{210}$Pb) as tracers of particle dynamics in a river system in southeast Michigan, Limnology and Oceanography, 53, 1934-1944, 2008.

Matisoff, G., Bonniwell, E. C., and Whiting, P. J.: Radionuclides as Indicators of Sediment Transport in Agricultural Watersheds that Drain to Lake Erie, Journal of Environmental Quality, 31, 62-72, 2002.

Matisoff, G., Wilson, C. G., and Whiting, P. J.: The $^{7}$Be/$^{210}$Pb$_{xs}$ ratio as an indicator of suspended sediment age or fraction new sediment in suspension, Earth Surf. Proc. Land., 30, 1191-1201, 2005.

Mudbidre, R., Baskaran, M., and Schweitzer, L.: Investigations of the partitioning and residence times of Po-210 and Pb-210 in a riverine system in Southeast Michigan USA. J. Environ. Radioact., 138, 375-383, 2014.

Saari, H. K., Schmidt, S., Castaing, P., Blanc, G., Sautour, B., Masson, O., and Cochran, J. K.: The particulate $^{7}$Be/$^{210}$Pb$_{xs}$ and $^{234}$Th/$^{210}$Pb$_{xs}$ activity ratios as tracers for tidal-to-seasonal particle dynamics in the Gironde estuary (France): implications for the budget of particle-associated contaminants, Sci. Total. Environ., 408, 4784-4794, 2010.

Wang, J., Du, J., Baskaran, M., and Zhang, J.: Mobile mud dynamics in the East China Sea elucidated using $^{210}$Pb, $^{137}$Cs, $^{7}$Be, and $^{234}$Th as tracers, J. Geophys. Res-Oceans, 121, 224-239, 2016.

Wang, J., Huang, D., Xie, W., He, Q., and Du, J.: Particle Dynamics in a Managed Navigation Channel Under Different Tidal Conditions as Determined Using Multiple Radionuclide Tracers, J. Geophys. Res-Oceans, 126, e2020JC016683, 2021.

Whiting, P. J., Matisoff, G., Fornes, W., and Soster, F. M.: Suspended sediment sources and transport distances in the Yellowstone River basin, Geol. Soc. Am. Bull., 117, 515-529, 2005.

Wieland, E., Santschi, P. H., and Beer, J.: A multitracer study of radionuclides in Lake Zurich, Switzerland: 2. Residence times, removal processes, and sediment focusing, J. Geophys. Res-Oceans, 96, 17067-17080, 1991.

P22, L450:   change "in areas" to "areas"

Response: The suggestion will be taken in the revised manuscript. "Concerning air concentrations in areas such as…" will be changed as "Concerning air concentrations, areas such as…"

P22, L454:   "which ARE almost"

Response: Thank you for noting this mistake – it is corrected in the revised manuscript.

P23, L470:   correct "SO4-".

Response: Thank you for noting this mistake –it is corrected in the revised manuscript.

P23, L471-472:     what is the connection between the 1st and 2nd sentences?

Response: Thank you for the suggestion. In order to make the text more connected and coherent, we have reorganized this paragraph (as below, move 1st sentence forward) "… quantification of the role of dry fallout in the removal of these nuclides will provide insights on the removal of other analog species. As mentioned earlier, combining cosmogenic $^{7}$Be with $^{210}$Pb which has a predominantly Earth-surface origin will be useful to trace species that originate both from Earth's surface, such as Hg, $SO_4^{2-}$, $NO_3^{-}$, and those that originate in the upper atmosphere, such as $O_3$. The size distribution of aerosols particles carrying $^{7}$Be and $^{210}$Pb is crucial for understanding atmospheric behavior and tracing analogues, and such studies also need to be conducted. Besides, the troposphere contains ~99% of global water vapor with < 1% in the stratosphere. The depositional velocity of aerosol in the stratosphere… "

P23, L473-474:   how about zonal transport?

Response: Thank you for the suggestion. This sentence is rephrased as "the $^{7}$Be concentration is governed by local production, zonal and vertical downward transport, and its decay".

P23, L474-477:   Do these lines mean the following?   "In the middle and upper troposphere where precipitation is much less frequent, the removal rate of aerosols is also slow. Collection of air samples in that part of the atmosphere will provide useful information on the total deposition velocity of aerosols (Lal and Baskaran, 2012)."

Response: Yes, our meaning here is consistent with the sentences you wrote above. We will replace these lines with the above sentences in the revised manuscript.

---

## Author Comment (AC2)

1 Response to Anonymous Reviewer #1

This manuscript provides an incredible contribution to the literature through the compilation of annual concentrations and annual deposition fluxes of Be-7 and Pb-210 around the world. Overall, the manuscript is well-written (although there are multiple typos throughout the text) and the data treatment/interpretation is of interest to a large audience (including vast research communities dealing with processes occurring in the atmosphere, the ocean, soils and rivers and for which the use of Be-7 and Pb-210 as a tracer is particularly useful).

9 We would like to thank the anonymous referee #1 for taking the time to provide a 10 thorough review of our submitted manuscript. The comments are very valuable and the 11 suggestions are very helpful. These comments and suggestions help us in greatly 12 improving the quality of our MS. In addition, language has been carefully further edited

- 13 by one of the coauthors Mark Baskaran.
- 14 Below, the original comments are in black, our responses are in blue.
- 15 General remarks
- 16 In my opinion, there is a research topic missing from the list, i.e. use of Be-7 and Pb-

210 as tracers of the sources and dynamics of riverine sediment (and not only soils and
ocean particles, there are transfers in-between both compartments). This should be

19 acknowledged in the text, with some supporting references.

Response: We fully agree this comment. We have incorporated the use of 7Be and 210Pb
as tracers for the sources and dynamics of sediments in freshwater systems (not only
rivers but also lakes). This is incorporated in the text throughout this paper. Specific
revisions are as follows:

 Abstract: 'for tracing soil redistribution processes on land and particle dynamics and...' will be changed as 'for tracing soil redistribution processes on land, particle dynamics in aquatic systems and mixing processes in open ocean...'

[revised manuscript text omitted]

Overall, I thought that there might be a confusion regarding Pb-210 measurements
between the supported Pb-210 and the unsupported Pb-210 (that referred to as 'excess
Pb-210'); could this be clarified in the text?

- 100 Response: This is clarified in section 2.2.3 as given below:
- 101  $^{210}$ Pbex is the difference between total (measured)  $^{210}$ Pb and the supported  $^{210}$ Pb in the
- soils. Supported 210Pb is assumed to be the same as 226Ra activity, under the assumption
- 103 of secular equilibrium between 226Ra and supported 210Pb. It can also be obtained by
- assuming that the supported 210Pb activity is equal to the total 210Pb at depth greater
- than 30 cm in the soil profile where atmospherically-delivered 210Pb has not reached
- 106 (Matisoff et al., 2014).
- 107 Reference
- Matisoff, G.: 210Pb as a tracer of soil erosion, sediment source area identification and particle transport
   in the terrestrial environment, J. Environ. Radioactiv., 138, 343-354, 2014.
- 110 Database
- 111 Regarding the dataset in itself, I am not sure that modifications can still be made, but I
- 112 wondered whether the monitoring period (from year x to year y, typically) could be
- added? Currently, to the best of my understanding, only the publication year is referred.

114 Response: The modifications of dataset can still be made. However, there may be some

115 misunderstanding here. The monitoring period (if available) has already been

**included in the dataset**. To alleviate the referee's concerns, we have attached a partial

screenshot (as below) of the dataset, **please note the part enclosed by the red frame.**

| 1   | A                                                                                                           | В                    | С             | D           | E           | F          | G                                  | Н                                          | 1                 | J        |                      |
|-----|-------------------------------------------------------------------------------------------------------------|----------------------|---------------|-------------|-------------|------------|------------------------------------|--------------------------------------------|-------------------|----------|----------------------|
| 1   | Site                                                                                                        | Sampling time        | Latitude (°N) | Longitude ( | Altitude (n | Annual pro | Sampling device                    | Filter                                     | Frequency         | Data nun | 7 Be annu |
| 148 | Jungfraujoch, Switzerla                                                                                     | Jul 1996-Dec 1998    | 46.53         | 7.98        | 3580        | NA         | air flow rate of 32-68 m3/h        | glass fiber (or cellulose nitrate) filters | 2 days            | 568      | 7.00                 |
| 149 | Jungfraujoch, Switzerla                                                                                     | Apr 1996-Jan 1997    | 46.53         | 7.98        | 3580        | NA         | high volume air samplers           | glass fiber (or cellulose nitrate) filters | 2 days            | ~120     | 6.80                 |
| 150 | Jungfraujoch, Switzerla                                                                                     | Mar 2000-Feb 2001    | 46.53         | 7.98        | 3580        | NA         | HIVOL air sampler with flow rat    | glass fibre filters                        | 2 days            | NA       | 5.60                 |
| 151 | Richland, USA                                                                                               | Jan 1967-Dec 1967    | 46.30         | -119.28     | NA          | NA         | NA                                 | NA                                         | NA                | NA       | 2.67                 |
| 152 | GERN, Switzerland                                                                                           | Jul 1998-Oct 2011    | 46.20         | 6.10        | 421         | NA         | ASS-500 sampler station with flo   | Petryanov filtering cloth                  | weekly            | NA       | 3.74                 |
| 153 | Wisconsin, USA                                                                                              | May 1994 and Aug 19  | 46.17         | -89.83      | NA          | NA         | Anderson high volume air sample    | quartz fiber filters                       | daily             | 43       | 4.00                 |
| 154 | Sondrio, Itlay                                                                                              | May 1991-April 1992  | 46.17         | 9.87        | 360         | NA         | electric blowing-fan (characterise | glass micro-fibre filters (diameter = 50   | daily             | NA       | 3.10                 |
| 155 | Monte Ceneri, Switzer                                                                                       | Jan 1994-Jun 1998 an | 46.10         | 8.90        | 586         | NA         | ASS-500 sampler station with flo   | Petryanov filtering cloth                  | weekly            | NA       | 3.94                 |
| 156 | 5 Ljubljana, Slovenia                                                                                       | Feb 2003-Dec 2011    | 46.09         | 14.59       | 281         | NA         | NA                                 | NA                                         | monthly           | 118      | 3.70                 |
| 157 | 7 Macugnaga, Milan, Itla                                                                                    | Feb 2011-Dec 2011    | 45.95         | 7.96        | 1300        | NA         | flow rate of 28.3 L/min            | Acetate Cellulose filters (0.8 µm pore     | quarterly         | 4        | 3.60                 |
| 158 | 3 Ispra, Milan, Itlay                                                                                       | Feb 2011-Dec 2011    | 45.82         | 8.61        | NA          | NA         | flow rate of 28.3 L/min            | Acetate Cellulose filters (0.8 µm pore     | quarterly         | 4        | 4.21                 |
| 159 | Brunate, Itlay                                                                                              | Oct 1992-May 1993    | 45.82         | 9.10        | 800         | NA         | blowing-fan (characterised by an   | glass micro-fibre filters (diameter 50 cr  | 2-3 days          | NA       | 2.10                 |
| 160 | Puy de Dome, France                                                                                         | Oct 2005-Jul 2008    | 45.77         | 2.97        | 1465        | NA         | high-volume sampler having a flo   | polypropylene fibres (Filters Jonell JP)   | biweekly          | ~80      | 4.23                 |
| 161 | Opme France                                                                                                 | Oct 2004-Jul 2008    | 45.72         | 3.07        | 660         | NA         | high-volume sampler having a flo   | polypropylene fibres (Filters Jonell JPM   | 10 days           | ~45      | 4.30                 |
| 162 | Beaverton, Oregon, US                                                                                       | Jan 1977-Dec 1985    | 45.53         | -122.88     | 64          | NA         | flow rate of about 1400 m3/day     | Microsorban air filter medium 99/97-4      | weekly            | 89       | 2.66                 |
| 163 | Beaverton, Oregon, U                                                                                        | Jan 1986-Dec 1993    | 45.53         | -122.88     | 64          | NA         | NA                                 | Dynaweb DW7301L filter material            | weekly            | 178      | 2.20                 |
| 164 | 1 Segrate, Milan, Itlay                                                                                     | Feb 2011-Dec 2011    | 45.49         | 9.29        | NA          | NA         | flow rate of 28.3 L/min            | Acetate Cellulose filters (0.8 µm pore     | quarterly         | 15       | 3.64                 |
| 165 | 5 Milano, Italy                                                                                             | Feb 1988-Jan 2011    | 45.47         | 9.18        | 125         | NA         | NA                                 | NA                                         | two weeks         | 473      | 3.00                 |
| 166 | 5 Milan, Itlay                                                                                              | Sept 1993-Jun 1995   | 45.47         | 9.17        | 120         | NA         | blowing-fan (characterised by an   | glass micro-fibre filters (diameter = 50   | daily             | NA       | 2.70                 |
| 167 | 7 University Degli Studi o                                                                                  | Feb 2011-Dec 2011    | 45.46         | 9.20        | NA          | NA         | flow rate of 28.3 L/min            | Acetate Cellulose filters (0.8 µm pore     | quarterly         | 3        | 3.59                 |
| 168 | B Hokkaido, Japan                                                                                           | Feb 2001-Aug 2001    | 45.32         | 142.17      | NA          | NA         | high volume air sampler (SIBAT.    | glass fiber filters (TOYO, GB-100R)        | weekly            | 19       | 2.60                 |
| 169 | Vinca, Serbia                                                                                               | May 2011-Sep 2012    | 44.89         | 20.60       | 95          | NA         | constant flow rate samplers (air f | Whatman 41, 15 cm×25 cm in diameter        | daily             | 15       | 5.06                 |
| 170 | Insitute, Belgrade, Serl                                                                                    | Apr 1994-Dec 2013    | 44.89         | 20.60       | 95          | 687.00     | Air samples were collected by co   | Whatman 41, 15 cm×25 cm in diameter        | daily             | 260      | 3.76                 |
| 171 | City, Belgrade, Serbia                                                                                      | Jan 2004-Apr 2009    | 44.78         | 20.53       | 205         | 700.00     | Constant flow rate samplers (ave   | FILTRAK/Whatman 41/DDR, 15 cm              | daily             | 52       | 2.73                 |
| 172 | Institute, Belgrade, Ser                                                                                    | Jan 2004-Apr 2009    | 44.89         | 20.60       | 95          | 700.00     | Constant flow rate samplers (ave   | FILTRAK/Whatman 41/DDR, 15 cm              | daily             | 52       | 2.54                 |
| 173 | Monaco                                                                                                      | Jan 1998-Dec 2010    | 44.83         | 7.50        | 15          | 622.00     | Sierra-Anderson (type 305-200      | Quartz microfiber filters of 0.8 µm por    | monthly           | 112      | 6.69                 |
| 174 | 1 Belgrade, Serbia                                                                                          | Jan 1996-Dec 2001    | 44.78         | 20.53       | 205         | 820.00     | flow rate of 25 m3/h               | FILTRAK/Whatman 41/DDR, 15 cm              | daily             | NA       | 2.10                 |
| 175 | Belgrade, Serbia                                                                                            | Jan 1991-Apr 1996    | 44.78         | 20.53       | 205         | 700.00     | flow rate of 20 m3/h               | FILTRAK/Whatman 41/DDR, 15 cm              | daily             | 64       | 4.04                 |
| 176 | Kumodraz, Belgrade,                                                                                         | Mar 2009-Dec 2011    | 44.74         | 20.51       | NA          | NA         | digital samplers DH 604EV.2 (F     | Cellulose filter paper FJ213340 1.770      | weekly            | ~140     | 1.76                 |
| 177 | 7 Bordeaux, France                                                                                          | NA (3 y)             | 44.70         | -0.70       | 30          | NA         | NA                                 | NA                                         | NA                | NA       | 3.49                 |
| 178 | Mt.Cimone, Italy                                                                                            | Jul 1996-Dec 1999    | 44.20         | 10.70       | 2165        | NA         | air flow rate of 32-68 m3/h        | glass fiber (or cellulose nitrate) filters | irregular interva | 264      | 5.30                 |
| 179 | Mt.Cimone, Italy                                                                                            | Jan 1998-Aug 2011    | 44.20         | 10.70       | 2165        | NA         | Thermo-Environmental PM10 hi       | rectangular glass fiber filters (Whatman   | weekly            | 1609     | 4.30                 |
| 180 | ) Ussuriysk, Russia                                                                                         | May 2009-Dec 2015    | 44.15         | 132.00      | 112         | NA         | NA                                 | NA                                         | daily             | NA       | 4.47                 |
|     | 7Pa annual concentration 210Dh ensuel concentration 7Pa consul descrition flux 210Dh ensuel descrition flux |                      |               |             |             |            |                                    |                                            |                   |          |                      |

In relation with this remark, how to explain the following statement: 'The dataset includes 494 annual surface air concentration data of 7Be covering 367 different sites, 366 annual surface air concentration data of 210Pb from 270 different sites, 304 annual depositional flux data of 7Be from 279 different sites, and 645 annual depositional flux data of 210Pb from 602 different sites.' >> these values at each site correspond to different years/periods then? I feel that this remains somewhat unclear...

Response: Yes, these values at each site correspond to different monitoring years/periods and were published in different articles. For example, at Malaga (Spain), the 7Be air concentration data during 1992-1995, 1996-2001, 2000-2006 and 2009-2012 were published in Dueñas et al. (1999), Dueñas et al. (2004), Dueñas et al. (2009) and Gordo et al. (2015), respectively.

- 129 Reference
- Dueñas, C., Fernández, M. C., Liger, E., and Carretero, J.: Gross alpha, gross beta activities and 7Be
   concentrations in surface air: analysis of their variations and prediction model, Atmos. Environ., 33,
   3705-3715, 1999.
- Dueñas, C., Fernández, M. C., Carretero, J., Liger, E., and Cañete, S.: Long-term variation of the
   concentrations of long-lived Rn descendants and cosmogenic 7Be and determination of the MRT of
   aerosols, Atmos. Environ., 38, 1291-1301, 2004.
- Dueñas, C., Fernández, M. C., Cañete, S., and Pérez, M.: 7Be to 210Pb concentration ratio in ground level
   air in Málaga (36.7°N, 4.5°W), Atmos. Res., 92, 49-57, 2009.
- Gordo, E., Liger, E., Dueñas, C., Fernandez, M. C., Canete, S., and Perez, M.: Study of 7Be and 210Pb as
   radiotracers of African intrusions in Malaga (Spain), J. Environ. Radioactiv., 148, 141-153, 2015.

Some of the results obtained in this meta-analysis are of very large interest for the 140 community. They could avoid colleagues to start monitoring Be-7 or Pb-210 fluxes and 141 rely on previous data monitoring. For instance, on Figure 7, providing the empirical 142 equations describing the relationships between annual precipitation and Be-7 143 depositional fluxes for different latitudinal bands would be extremely useful (at least 144 for those latitudinal bands where the relationship is satisfactory) >> could they be added 145 in a table and made accessible to the community? The same suggestion could be made 146 for Pb-210 in Figure 8. 147

Response: Thank you for the suggestion. The empirical equations describing the
 relationships between annual precipitation and depositional fluxes of 7Be and 210Pb for
 different latitudinal bands have been added in a table as given below. The Pearson's r,

- 151 p-value and number of data points have also been added in Table 2.
- 152 Table 2. A summary of empirical equations and fitting parameters describing the
- relationships between annual precipitation (x) and 7Be and 210Pb depositional fluxes (y) for different latitudinal bands

| Nuclides          | Latitudinal band | Empirical equation | Pearson's r | p-value | Number of points |
|-------------------|------------------|--------------------|-------------|---------|------------------|
|                   | 60°N-70°N        | y=2.97x-1000.3     | 0.89        | 1.1E-1  | 4                |
|                   | 50°N-60°N        | y=2.16x-540.0      | 0.95        | 7.2E-7  | 13               |
|                   | 40°N-50°N        | y=1.71x+183.4      | 0.76        | 3.8E-9  | 43               |
|                   | 30°N-40°N        | y=1.40x+97.5       | 0.64        | 1.3E-8  | 64               |
| 7 Be   | 20°N-30°N        | y=0.29x+653.9      | 0.38        | 9.1E-2  | 21               |
|                   | 10°N-20°N        | y=0.54x+297.9      | 0.95        | 5.3E-2  | 4                |
|                   | 10°S-20°S        | y=0.76x+293.8      | 0.67        | 2.1E-1  | 5                |
|                   | 20°S-30°S        | y=1.50x+302.5      | 0.80        | 2.0E-1  | 4                |
|                   | 30°S-40°S        | y=2.52x-297.4      | 0.99        | 1.9E-2  | 3                |
|                   | 70°N-80°N        | y=0.04x+0.07       | 0.84        | 6.8E-4  | 12               |
|                   | 60°N-70°N        | y=0.10x-16.1       | 0.76        | 4.2E-5  | 22               |
|                   | 50°N-60°N        | y=0.03x+74.9       | 0.25        | 2.5E-2  | 31               |
|                   | 40°N-50°N        | y=0.06x+117.5      | 0.25        | 2.7E-4  | 206              |
|                   | 30°N-40°N        | y=0.13x+71.8       | 0.39        | 2.1E-5  | 113              |
|                   | 20°N-30°N        | y=0.25x-124.6      | 0.59        | 1.8E-7  | 67               |
|                   | 10°N-20°N        | y=0.09x-6.4        | 0.94        | 1.5E-3  | 7                |
| 210 Pb | 0°N-10°N         | y=-0.03x+239.9     | 0.29        | 5.3E-1  | 7                |
|                   | 10°S-20°S        | y=0.06x-2.2        | 0.66        | 5.3E-2  | 9                |
|                   | 20°S-30°S        | y=0.01x+56.5       | 0.15        | 6.5E-1  | 11               |
|                   | 30°S-40°S        | y=0.11x-31.3       | 0.65        | 3.7E-3  | 18               |
|                   | 40°S-50°S        | y=0.06x-3.5        | 0.80        | 1.8E-3  | 12               |
|                   | 60°S-70°S        | y=0.01x+1.7        | 0.77        | 9.4E-3  | 10               |
|                   | 70°S-80°S        | y=0.02x+0.2        | 0.86        | 6.1E-3  | 8                |
|                   | 80°S-90°S        | y=0.02x+0.5        | 0.92        | 2.1E-4  | 10               |

A similar remark can be made regarding Fig. 9: how could this very useful data compilation on Be-7/210Pb activity ratios be of further use for the community in the future? Could the range in ratios found in different latitudinal bands be provided somewhere (e.g. in a table)?

- Response: Thank you for the suggestion. We have uploaded the 7Be/210Pb concentration
  ratio and flux ratio data (regarding Fig. 9) in the dataset. Furthermore, we have also
  uploaded the deposition velocities (Vd) data for aerosols calculated from 7Be and 210Pb
  (Fig. 10) in the dataset. A new DOI (https://doi.org/10.5281/zenodo.4785136) of new
  version dataset is provided in the revised manuscript.
- A final general question (that could be addressed in section 3.7 for instance) is to think about the potential inclusion of nuclear safety continuous monitoring data (e.g. those monitored by state agencies in charge of nuclear safety) in future global databases, what would be the opinion of the authors on that?
- Response: This is an interesting proposition. The inclusion of nuclear safety continuous 168 monitoring data in future global databases will undoubtedly fill some gaps and expand 169 the scope of this dataset. However, this involves an important issue: data sharing. Data 170 sharing is a valuable part of the scientific method allowing for verification of results 171 and extending research from prior results. Scientific data are not only the outputs of 172 research but provide inputs to new hypotheses, enabling new scientific insights and 173 driving innovation. However, barriers to effective data sharing and preservation are 174 deeply rooted in the practices and culture of the research process as well as the 175 176 researchers themselves (Tenopir et al., 2011). During our compilation of this dataset, we encountered some obstacles. Some data is difficult to obtain directly from the 177 literature (whether from text or figures), so we contacted the authors, but sometimes 178 did not receive a reply. For some old data, the author cannot even be contacted. Of 179 course, we also received some generous and friendly helps. Thus, there is still some 180 181 data not included in our dataset, although we have tried our best. Finally, back to the question itself, we believe that the inclusion of nuclear safety continuous monitoring 182 data in future global databases requires more extensive collaboration and data sharing. 183 Hope our dataset can be a starting point. 184
- 185 Reference
- Tenopir, C., Allard, S., Douglass, K., Aydinoglu, A.U., Wu, L., Read, E., Manoff, M., and Frame, M.:
  Data sharing by scientists: practices and perceptions, PLoS ONE, 6, e21101, http://doi.org/10.1371/journal.pone.0021101, 2011.
- 189 Detailed remarks throughout the text
- 190 Abstract

L.17 "for tracing soil redistribution processes on land and particle dynamics and mixing
processes in the ocean" >> Be-7 and Pb-210 are also widely used for quantifying the
sources and the dynamics of riverine sediment (not only soils or ocean particles as
mentioned in the current version of the text)

195 Response: As given above, this missing research topic is now included in the text 196 throughout this paper; the words 'in aquatic systems' is be added after 'particle 197 dynamics'

- 198 L.21 I would remove the second 'of'
- 199 Response: The second 'of' is removed in the revised version.

L.25 'future researchers' public consumption in their research' >> unclear what is
 meant here

202 Response: Here we mean that the dataset is freely available for the scientific community. sentence will be rephrased as 'The dataset is archived This 203 https://doi.org/10.5281/zenodo.4785136 (Zhang et al., 2021) and is freely available for 204 the scientific community. The purpose of this paper is to provide an overview of the 205 scope and nature of this dataset and its potential utility as baseline data for future 206 research. 207

- 208 Introduction
- 209 L.29 Earth's surface > Earth' surface
- 210 Response: Thank you for noting this mistake it is corrected it in the revised manuscript.
- 211 Similar mistakes throughout the text are also corrected.
- 212 L.32 they do not >> it does not?
- 213 Response: Thank you for noting this mistake –it is corrected in the revised manuscript.
- 214 L.33 and changing >> which changes?
- 215 Response: This sentence is rewritten as A major fraction of  $^{7}\text{Be}$  (67%) production takes

216 place in the stratosphere, but it does not readily reach the troposphere except during

- 217 spring when seasonal thinning of tropopause folds near the jet stream take occurs at 218 mid-latitudes'.
- L.40 while not providing a range of Rn-222 fluxes for the oceanic areas as for the continental fluxes?
- 221 Response: We have added this in the revised version: "Rn-222 fluxes for the oceanic
- areas ranged from 2 to 21 Bq m-2 d-1 (Wilkening and Clements, 1975)."
- 223 Reference
- Wilkening, M. H., and Clements, W. E.: Radon 222 from the ocean surface, J. Geophys. Res., 80, 38283830, 1975.
- L.41 a part of the sentence is missing here (at the end of L.41?)
- Response: Thank you for noting this mistake This sentence is deleted in the revised
  manuscript.
- 229 L.49 'in accumulation mode'? >> unclear what is meant here
- 230 Response: Atmospheric aerosols are typically described as consisting of three modes
- based on their sizes: the nucleation mode (0.01-0.1 µm), accumulation mode (0.1-1.0
- 232  $\mu$ m), and coarse mode (> 1  $\mu$ m) (Whitby, 1978; Meng and Seinfeld, 1994). The size of
- 233 aerosol particles determines to a large extent how they are transported and transformed

234 in the atmosphere and how they are removed. Accumulation mode aerosol particles are

- removed from atmosphere primarily by precipitation because they are too small for
- 236 gravitational settling and removal and too large to be deposited by Brownian motion.
- 237 Reference
- 238 Whitby, K. T.: The physical characteristics of sulfur aerosols, Atmos. Environ., 12, 135-159, 1978.
- Meng, Z. Y., and Seinfeld, J. H.: On the source of the submicrometer droplet mode of urban and regional
   aerosols, Aerosol Sci. Tech., 20, 253-265, 1994.

241 L.54 and similar tropospheric...?

242 Response: The suggestion is taken into consideration in the revised manuscript.

L.66 (and elsewhere); of note, this type of research is also widely conducted in
freshwater/ river environments and could be acknowledged in the text, e.g.

Response: Thank you for the suggestion. As mentioned in the response above,
'Meanwhile, 7Be and 210Pb are also widely used for indicating particle transport,
deposition, and resuspension in estuarine and coastal regions' is rewritten as
'Meanwhile, 7Be and 210Pb are also widely used as tracers of sediment source
identification and particle dynamics in rivers (e.g. Bonniwell et al., 1999; Matisoff et
al., 2005; Jweda et al., 2008; Mudbidre et al., 2014; Baskaran et al., 2020), lakes (e.g.
Dominik et al., 1987; Schuler et al., 1991; Vogler et al., 1996), estuaries and coasts (e.g.

- 252 Baskaran et al., 1997; Huang et al., 2013; Wang et al., 2016).' in the revised manuscript.
- 253 Reference
- Baskaran, M., Ravichandran, M., and Bianchi, T. S.: Cycling of 7Be and 210Pb in a High DOC,
  Shallow, Turbid Estuary of South-east Texas, Estuar. Coast. Shelf S., 45, 165-176, 1997.
- Baskaran, M., Mudbidre, R., and Schweitzer, L.: Quantification of Po-210 and Pb-210 as tracer of
  sediment resuspension rate in a shallow riverine system: case study from Southeast Michigan,
  USA, J. Environ. Radioact., 222, http://doi.org/10.1016/j.jenvrad.2020.106339, 2020.
- Bonniwell, E. C., Matisoff, G., and Whiting, P. J.: Determining the times and distances of particle
  transit in a mountain stream using fallout radionuclides, Geomorphology, 27, 75-92, 1999.
- Dominik, J., Burrus, D., and Vernet, J. P.: Transport of the environmental radionuclides in an alpine
   watershed, Earth Planet. Sc. Lett., 84, 165-180, 1987.
- Huang, D., Du, J., Moore, W. S., and Zhang, J.: Particle dynamics of the Changjiang Estuary and
   adjacent coastal region determined by natural particle-reactive radionuclides (7Be, 210Pb, and
   234Th), J. Geophys. Res-Oceans, 118, 1736-1748, 2013.
- Jweda, J., Baskaran, M., van Hees, E., and Schweitzer, L.: Short-lived radionuclides (7Be and 210Pb)
  as tracers of particle dynamics in a river system in southeast Michigan, Limnology and
  Oceanography, 53, 1934-1944, 2008.
- Matisoff, G., Wilson, C. G., and Whiting, P. J.: The 7Be/210Pbxs ratio as an indicator of suspended
  sediment age or fraction new sediment in suspension, Earth Surf. Proc. Land., 30, 1191-1201,
  2005.
- Mudbidre, R., Baskaran, M., and Schweitzer, L.: Investigations of the partitioning and residence
  times of Po-210 and Pb-210 in a riverine system in Southeast Michigan USA. J. Environ.
  Radioact., 138, 375-383, 2014.

- Schuler, C., Wieland, E., Santschi, P. H., Sturm, M., Lueck, A., Bollhalder, S., Beer, J., Bonani, G.,
  Hofmann, H. J., Suter, M., and Wolfli, W.: A multitracer study of radionuclides in Lake Zurich,
  Switzerland: 1. Comparison of atmospheric and sedimentary fluxes of 7Be, 10Be, 210Pb, 210Po,
  and 137Cs, J. Geophys. Res., 96, 17051-17065, 1991.
- Vogler, S., Jung, M., and Mangini, A.: Scavenging of 234Th and 7Be in Lake Constance, Limnol.
  Oceanogr., 41, 1384-1393, 1996.
- Wang, J., Du, J., Baskaran, M., and Zhang, J.: Mobile mud dynamics in the East China Sea
  elucidated using 210Pb, 137Cs, 7Be, and 234Th as tracers, J. Geophys. Res-Oceans, 121, 224-239,
  2016.
- 284 L.77 IMS operated by CTBTO?
- 285 Response: The suggestion is incorporated in the revised manuscript.
- 286 Methods
- 287 L.89 'high volume air' >> a high volume of air?
- 288 Response: The suggestion is incorporated in the revised manuscript
- 289 L.101 spectrometry instead of spectroscopy?
- Response: Thank you for noting this mistake we have corrected it in the revisedmanuscript.
- 292 L.111 'tedious procedures' >> unclear what is meant here

Response: The 'tedious procedures' refers to the continuous and tedious measurement
(preclean of rain collectors, preconcentration of rain samples, determination of
chemical yield, etc.) of the 7Be and 210Pb concentration in precipitation. We also note a
mistake in this sentence – the word 'avoids' is missing here. The sentence is rephrased
as: 'use of natural archives avoids the labor and time-intensive measurements of 7Be
and 210Pb concentration in precipitation and can serve as a complement...' in the revised
manuscript.

- 300 L.112 'deserted areas' >> unclear what is meant here
- 301 Response: 'deserted areas' here refer to areas where continuous monitoring is difficult
- 302 (such as open ocean, alpine region and polar region). To avoid misunderstanding, we 303 changed it to: 'remote areas' in the revised manuscript.
- 304 L.114 'to an undisturbed area' > to undisturbed areas?
- 305 Response: The suggestion is incorporated in the revised manuscript.
- 306 L.123 yields > yield?
- 307 Response: Thank you for noting this mistake corrected it in the revised manuscript.
- 308 L.133 'immediately after' >> immediately added after?
- 309 Response: The suggestion is incorporated in the revised manuscript.

- 310 L.137 'was not done resulting in underestimate of depositional' >>resulting in the
- 311 underestimation of...?
- 312 Response: The suggestion is incorporated in the revised manuscript.
- 313 L.149 After deposited >> after being deposited?
- 314 Response: Suggested revision is made.
- 315 LL.150-51: 'Open Ocean' >> why using capital letters here (instead of open ocean)?
- Response: Thank you for noting this mistake we have corrected in the revisedmanuscript.
- 318 L.169 have shown that the atmospheric fluxes
- 319 Response: The suggestion is incorporated in the revised manuscript.
- 320 L.171 "and hence those are data are not included" >> unclear, please rephrase
- 321 Response: This sentence is rephrased as: "the data of 7Be soil inventory are not included
- 322 in our data set" in the revised manuscript.
- L.177 'sediment focusing and erosion" >> unclear what is referred to with 'sedimentfocusing'
- Response: In lake basin, surficial finer sediment may be resuspended due to bottom currents and and/or tidal currents in shallow water and subsequently transported to areas to specific areas which are conducive to deposition, in particular, especially during overturn (Davis, 1968). This phenomenon, which results in redistribution of bottom sediments resulting in higher accumulation in certain areas of the lake/estuaries/coastal areas which results in areas of sediment focusing (Likens and Davis, 1975).
- 331 References
- Davis, M. B.: Pollen grains in lake sediment, redeposition caused by seasonal water circulation, Science,
   162, 796-799, 1968.
- Likens, G. E. and Davis, M. B.: Post-glacial history of Mirror Lake and its watershed in New Hampshire,
   USA: An initial report, Int. Ver. Theor. Angew. Limnol. Vcrh, 19, 982-993, 1975.
- LL.182-184 'one is generated from the decay of 222Rn in the soil minerals, known as supported 210Pb which is produced from the decay of 238U and the other comes from atmospheric deposition as unsupported 210Pb. The fallout of 210Pb is retained generally in the organic rich surface soils presumably because of the sequestering properties of the organic matter as well as in lithogenic mineral grain.' >> this seems to reflect the old vision that there are a mineral and an organic component in soils, instead of the occurrence of 'organo-mineral complexes'
- Response: Thank you for suggestion. The sentence is rephrased in the revised manuscript as 'The fallout of 210Pb is retained generally in the organic rich surface soils presumably because of the sequestering properties of the organo-mineral complexes (Covelo et al., 2008)'.

- L.187 'concentration than that expected' >> higher than that/compared to that...?
- 351 Response: The suggestion is taken into consideration in the revised manuscript.
- 352 L.197 'at different sampling time' >> sampling times
- 353 Response: Revision is made.
- L.200 'possibility of the dating ice core' >> 'possibility of dating ice cores'?
- 355 Response: The suggestion is incorporated in the revised manuscript.
- 356 L.203 and the Arctic?
- 357 Response: The suggestion incorporated in the revised manuscript.
- 358 L.204 'small montane permanent snow filed' >> unclear what is meant here (maybe 359 snowfield...)?
- Response: Thank you for noting this mistake 'snow filed' here is corrected to 'snowfield' in the revised manuscript.
- 362 L.205 'in the same way as the soil' >> in the same way as for the soil, except that...?
- 363 Response: The suggestion is incorporated in the revised manuscript.
- 364 L.208 'are very low' > is very low?
- 365 Response: Thank you for noting this mistake it is corrected it in the revised manuscript.
- 366 L.214 'Regarding compiling' >> please rephrase
- Response: We have replaced: 'Regarding compiling the global dataset for annual 7Be and 210Pb air concentrations and depositional fluxes' with: 'In order to compile the global dataset for annual 7Be and 210Pb air concentrations and depositional fluxes comprehensively' in the revised manuscript.
- 371 L.226 was included > were included?
- 372 Response: Thank you for noting this mistake it is corrected in the revised manuscript
- 373 L.228 'originating authors' > unclear, I would rephrase this
- 374 Response: 'the originating authors and editors have taken...' is rephrased as 'the
- authors and editors of the original articles have taken...' in the revised manuscript.
- 376 LL.229-230 convert in >> convert into?
- 377 Response: The suggestion is incorporated in the revised manuscript.
- 378 L.234 'program' >> which program is referred to here?

Response: The program refers to GetData Graph Digitizer. We has added thisinformation in the revised manuscript.

381 LL.235-236 'In rare cases, only the locality name of the study site was available, the 382 geographical location was digitized by Google Earth.' >> unclear here, do you mean 383 that the approximate coordinates were extracted from Google Earth?

Response: Yes, the approximate coordinates were extracted from Google Earth. To alleviate the referee's concern, 'the geographical location was digitized by Google Earth' will be rephrased as 'the geographical coordinates were extracted from Google

- 387 Earth' in the revised manuscript.
- 388 Results and discussion
- 389 L.247 in different literature >> unclear what is meant here
- 390 Response: To avoid misunderstanding, the 'literature' here is changed to 'articles' in
- 391 the revised manuscript.
- 392 Figure 1: the a/b/c/d letters referring to the different figure panels are not easy to see,
- 393 could there be a way to make them visible?

Response: The figure 1 has been replotted (as below) to make the a/b/c/d letters more visible.

- 1255 'A number' > the number?
- 397 Response: The suggestion is incorporated in the revised manuscript.
- 398 L.257 'earlier than that' > those?
- Response: Thank you for noting this mistake we have corrected it in the manuscript.
- 400 L.259 work was started >> I would remove 'was'?

- Response: The suggestion is incorporated in the revised manuscript. 401
- L.271 'in the undisturbed site' >> in an undisturbed site? 402
- Response: The suggestion is incorporated in the revised manuscript. 403
- 404 L.284 mainly dedicated to investigate...
- Response: The suggestion is incorporated in the revised manuscript. 405
- L.285 Be-7 >> are you referring to the Be-7 fluxes here? 406
- Response: No, we are referring to the 7Be air concentrations and depositional fluxes. 407
- L.295 I would refer to the concentrations and depositional fluxes separately in the 408 sentence to facilitate its reading 409
- Response: The suggestion is incorporated in the revised manuscript. The sentence will 410
- be rephrased as "The range of concentrations of 7Be and 210Pb are 0.33-17.77 mBq m-3 411
- and 0.003-4.65 mBq m-3, respectively. The range of depositional fluxes of 7Be and 210Pb 412
- are 59-6350 Bq m-2 y-1 and 1-2539 Bq m-2 y-1, respectively." in the revised manuscript. 413
- L.331 for Pb-210 than for Be-7 414
- Response: Thank you for noting this mistake we have incorporated this in the revised 415
- manuscript 416
- L.332 'However' >> why starting the sentence with 'however'? 417
- Response: Thank you for noting this mistake 'however' is deleted in the revised 418 manuscript 419
- 420 Figure 6 – caption – L. 338: 'against with' >> versus?
- Response: The suggestion is incorporated in the revised manuscript. 421
- LL.342-43 'less than 5% of that in the same latitude' >> unclear what is meant here? 422
- Response: 'less than 5% of that in the same latitude' will be rephrased as 'less than 5% 423 of the global average 7Be flux' in the revised manuscript. 424
- L.345 'Hokitika' >> I don't know this location, where is it located? 425
- Response: 'Hokitika' is located in New Zealand, we have added this information in the 426 revised manuscript. 427
- Figure 8 caption L. 358: latitudinal bands (in plural)? (same remark in Fig. 7) 428
- Response: Thank you for noting this mistake we have corrected this in the revised 429 manuscript 430
- L.368 in 19 sites for which (...) ratios were available,...? 431
- Response: The suggestion is incorporated in the revised manuscript. 432

- 433 L.368 the paired t-test > a paired t-test?
- 434 Response: The suggestion is incorporated in the revised manuscript.
- L.375 'their measurements are easy' >> this is all relative, depending on the point ofview...
- 437 Response: We have deleted this sentence in the revised manuscript.
- 438 L.389 'is an artifact of the manner in the calculation' >> in the calculation mode?
- 439 Response: The suggestion is incorporated in the revised manuscript.
- 440 L.405 were used > was used?
- Response: Thank you for noting this mistake, we have corrected it in the revisedmanuscript
- 443 L.418 particle dynamics > riverine particle dynamics?
- 444 Response: Thank you for the suggestion. Considering that 7Be and 210Pb are also widely
- 445 used as tracers of sediment source identification and particle dynamics not only in rivers,
- 446 but also in lakes, estuaries and coasts, we believe that it is more appropriate to use

447 'aquatic particle dynamics' here. Thus, 'particle dynamics' is changed to 'aquatic

- 448 particle dynamics' in the revised manuscript.
- 449 Section 3.6: As mentioned above, I think that riverine particle dynamics using Be-7 and450 Pb-210 measurements should be addressed in this section.
- Response: Thank you for suggestion. As in the response to general remarks above, the
  riverine particle dynamics using 7Be and 210Pb measurements is addressed in this
  section:
- 'In the estuarine and coastal areas, the mass balance calculations of 7Be and...' is 454 rephrased as 'In aquatic systems (including river, lake, estuary and coast), the mass 455 balance models of 7Be and 210Pbex have become powerful tools to understand the 456 sediment source, transportation and resuspension processes (e.g. Wieland et al., 1991; 457 Feng et al., 1999; Jweda et al., 2008; Huang et al., 2013; Mudbidre et al., 2014), in such 458 models, the atmospheric depositional input of 7Be and 210Pb is a required source term. 459 In addition, 7Be/210Pbex activity ratio can be used to identify the source area of sediments 460 (Whiting et al., 2005; Jweda et al., 2008; Wang et al., 2021), to quantify the age of 461 sediments (Matisoff et al., 2005; Saari et al., 2010), and to determine the transport 462 distance of suspended particles (Bonniwell et al., 1999, Matisoff et al., 2002). Thus, the 463 atmospheric depositional flux data of 7Be and 210Pb are also important for tracing 464 particle dynamics in aquatic systems' 465
- 466 L.423 of an undisturbed > at an undisturbed?
- 467 Response: The suggestion is incorporated in the revised manuscript.
- 468 L.425 'exceeding' > enrichment?
- 469 Response: The suggestion is incorporated in the revised manuscript.

- 470 L.425 'accumulation and/or redistribution' >> unclear which difference you make
- 471 between both processes here?
- 472 Response: We deleted 'and/or redistribution' in the revised manuscript.
- 473 L.432 'indicates notable sediment focusing or additional particle input other than
  474 atmospheric fallout' >> unclear what is meant here, please rephrase
- 475 Response: Due to the extensive modification of the section 3.6, this sentence is deleted
- 476 in the revised manuscript.
- LL.443-444: '7Be depositional flux is independent of longitude and is constant over
  broad latitudinal bands. Thus, the 7Be depositional flux data in our dataset can be used
  to estimate 7Be ocean inventory in the same latitude, which can avoid the collection of
  the large volume of seawater samples and extend the application of 7Be in the Open
  Ocean' >> I fully agree with the authors here and I think that this could be further
  outlined in the text (including for continental locations)
- 483 Response: Thank you for the suggestion. We will add a new paragraph (as below) in484 section 3.6 to for clarification:
- 485 Scientific data are not only the outputs of research but also provide inputs to new hypotheses, extending research and enabling new scientific insights (Tenopir et al., 486 2011). Our dataset provides a forum in which a large amount of 7Be and 210Pb 487 atmospheric depositional flux data for the above-mentioned research communities. 488 Researchers can rely on previously collected data in planning their research, without 489 additional monitoring of 7Be and/or 210Pb depositional fluxes. Even for those areas with 490 data gaps, the empirical equations between 7Be and 210Pb depositional fluxes and annual 491 precipitation (Table 2) provide an empirical method for estimating fluxes, especially 492 for 7Be, as 7Be depositional flux is independent of longitude and is constant over broad 493 latitudinal bands. In summary, the atmospheric depositional flux data presented in our 494 dataset as well as the meta-analysis of the data will be useful in the investigations of 495 soil erosion studies in terrestrial environments, particle dynamics studies in aquatic 496 systems, and surface mixing process studies in open ocean. 497
- 498 Reference
- Tenopir, C., Allard, S., Douglass, K., Aydinoglu, A.U., Wu, L., Read, E., Manoff, M., and Frame, M.:
  Data sharing by scientists: practices and perceptions, PLoS ONE, 6, e21101, http://doi.org/10.1371/journal.pone.0021101, 2011.
- 502 L.454 are almost non-existent
- Response: Thank you for noting this mistake we have corrected it in the revisedmanuscript.
- 505 L.468 meteorological conditions?
- 506 Response: The suggestion is incorporated in the revised manuscript.
- 507 L.481 'from the same literature' >> article?

- 508 Response: This suggestion is incorporated in the revised manuscript.
- 509 Conclusions
- 510 L.486 'spanning the time from 1955 to early 2020' >> spanning the period...?
- 511 Response: The suggestion is incorporated in the revised manuscript.
- 512 L.493 may be add 'in river systems' after dynamics here?
- 513 Response: 'in aquatic systems' is added here in the revised manuscript.